



# Assessment of the wind energy resource on the coast of China based on machine learning algorithms

Boming Liu[1], Xin Ma[1], Jianping Guo[2*], Hui Li[1], Shikuan Jin[1], Yingying Ma[1], and Wei Gong[1]

[1] State Key Laboratory of Information Engineering in Surveying, Mapping and Remote Sensing (LIESMARS), Wuhan University, Wuhan, China

[2] State Key Laboratory of Severe Weather, Chinese Academy of Meteorological Sciences, Beijing 100081, China

*Correspondence to*: Dr./Prof. Jianping Guo (Email: jpguocams@gmail.com)

**Abstract.** Wind is one of the most essential clean and renewable energy sources in today's world. To achieve the goal of carbon emission peak and carbon neutrality in China, it is necessary to evaluate the wind energy resources on the coast of China. Nevertheless, the traditional power law method (PLM) relies on the constant coefficient to estimate the high-altitude wind speed. The constant assumption may lead to significant uncertainties in wind energy assessment, given the large dependence on a variety of factors. To minimize the uncertainties, we here use three machine learning (ML) algorithms to estimate high-altitude wind from surface wind. The radar wind profiler and surface synoptic observations at eight coastal stations from May 2018 to August 2020 are used as key inputs to investigate the wind energy resource. Afterwards, three ML and the PLM are used to retrieve the wind speed at 120 m above ground level ($WS_{120}$). The comparison results show the random forest (RF) is the most suitable model for the estimation of $WS_{120}$. As such, the diurnal variation of $WS_{120}$ and wind power density (WPD) are then evaluated based on the $WS_{120}$ from RF model. For land stations, the hourly mean WPD is larger at daytime from 0900 to 1600 local solar time (LST) and reach a peak at 1400 LST. This is mainly due to the influence of the prevailing sea breeze. On the contrary, the hourly mean WPD of island stations is relatively large at nighttime during 1800 to 2300 LST. This indicates that the wind energy peaks differ by the land surface types. In terms of the spatial distribution of the seasonal mean $WS_{120}$ and WPD along the coastal region of China, the WPDs at Qingdao, Dayang, and Dongtou are higher than 200 $W/m^2$ in most seasons, and the WPDs at Dongying, Penglai, Qingdao, and Lianyungang are much greater than at Fuqing and Zhuhai. The result shows that the coastal regions of Bohai Sea and Yellow Sea have more abundant wind resources than those of East China Sea and the South China Sea. These findings obtained here provide insights into the development and utilization of wind energy industry on the coast of China in the future.

**Key words:** wind energy, radar wind profiler, remote sensing, machine learning





## 1. Introduction

With the rapid economic development of the world, the massive consumption of fossil fuels produces an increasing amount of carbon dioxide emission (Shi et al., 2021; Pei et al., 2022). Due to the ever-
rising air temperature and the depletion of fossil energy, it is increasingly becoming imperative to develop renewable clean energy (Hong et al., 2012). Among the myriad renewable energy resources, wind energy has gained more and more favors because of its abundant availability, good sustainability, and cost-effectiveness (Li et al., 2018), showing great promising prospect of commercial application (Leung et al., 2012). By the end of 2020, the global cumulative installed capacity had reached 743 GW
(Global Wind Report, 2021). It is estimated that wind power will account for approximately one-third of the increase in renewable power generation by 2035 (Khatib, 2012). Therefore, accurate estimation of wind profile is of great importance.

In recent decades, wind energy has been extensively studied all over the world. The 2009 edition of Wind Energy Facts elaborated on all aspects of wind energy in Europe at full length (EWEA, 2009).
Durisic et al. (2012) used wind data measured at four different heights to analyze the wind power in the South Banat region. Li et al. (2018) analyzed the wind speed and compared wind energy resources at offshore, nearshore, and onshore locations near Lake Erie. Their results showed that the offshore stations can offer more wind energy than onshore stations. Oh et al. (2012) applied the wind speed and wind direction data recorded in three meteorological masts to assess the wind energy and predict the
annual energy production at the demonstration offshore wind farm in Korea. Based on 17 years of wind data on Deokjeok-do island, Ali et al. (2018) investigated the wind characteristics during different time scale. Band et al. (2021) estimated the wind energy in the Gulf of Oman by using the near-surface wind data from the Middle East and North Africa-COordinated Regional Downscaling Experiment. As one of the largest energy consuming counties in the world, China is currently confronted with an
increasingly serious energy and climate situation (Khatib et al., 2012). The Chinese government proposes the carbon emission peak and carbon neutrality strategy to deal with energy and environmental issues (Pei et al., 2022). With the stimulus of policies and the favor of investors, wind power industry in China has been flourished. It is reported that the top market of the world by the end of 2020 for cumulative wind power installations was China (Global Wind Report, 2021). The wind is
well recognized to be produced by the difference of atmospheric pressure gradient (Solanki et al., 2022), which is deeply affected by the factors such as inhomogeneous underlying surface, land sea difference and ubiquitous atmospheric turbulence (Tieleman 1992; Coleman et al., 2021). As such, vertically resolved wind varies constantly by time and space, thereby making accurate wind energy assessment to be a big challenge to date.



Wind turbine is generally installed at the top of wind mast with a height of 100-120 m above ground
level (AGL), which roughly corresponds to the surface layer. The surface layer is most susceptible to
the influence of Earth's surface processes and tends to develop to an altitude at one tenth of the PBL
height during daytime (Li et al., 2021). It follows that Monin–Obukhov similarity theory does not work
for most times of a day, particularly over heterogenous land surface like costal region and in the
presence of convective cloud (Maronga and Reuder, 2017). At present, the traditional measurement of
wind usually uses meteorological masts equipped with anemometers, wind vane and other devices
(Shu et al., 2016). It is noted that the anemometer is usually installed at 10 m AGL, while the shoreline
wind turbine is usually installed at 100-120 m AGL. The wind speed at 100-120 m AGL was
commonly calculated by the power law method (PLM) (Hellman et al. 1914). The PLM method
generally assumes the wind speed below 150 m in the planetary boundary layer (PBL) varies
exponentially with height. Nevertheless, due to the ubiquitous nature of turbulence in the PBL, wind
varies constantly and greatly in the vertical (Stull 1988; Solanki et al., 2022), posing great challenges
to wind estimate. The significant impact of heterogeneous surface roughness on wind profiles further
complicates this issue (Tieleman 1992; Liu et al., 2018; Coleman et al., 2021). It leads to large
uncertainties in estimating wind at high altitude from the anemometer measurement.

The vertical wind profiles can also be observed by instruments such as radiosondes (RS) and radar
wind profiler (RWP). The RS can measure the wind speed at different heights in real time during the
rising process, which is one of the means to monitor the high-altitude wind energy (Li et al., 2021).
One noteworthy drawback is that the operational RS only provide wind profiles twice per day: 0800
and 2000 local solar time (LST). In contrast, active remote sensing equipment, such as RWP, can
measure the temporal variation of wind profiles, starting from the ground surface up to a height of 5-
8 km AGLL (Liu et al., 2019; Guo et al., 2021a). Nevertheless, there exists large uncertainties in the
wind speed near the ground surface measured by the RWP due to the influence of surface clutter.

Given the abovementioned problems, we attempt to use machine learning (ML) algorithm to retrieve
wind speed at 120 m AGL ($WS_{120}$). The surface in situ wind speed, high-altitude RWP wind speed and
corresponding surface meteorological data from May 2018 to August 2020 are collected to develop
the ML models. The performance of classical PLM method and three ML models were then compared.
Next, the most effective model was used to assess the wind power on coast of China. The results of
our study can provide useful information for the development of wind energy industry on the coast of
China. The observational data is briefly introduced in section 2. The ML model construction and wind



energy evaluation method are displayed in section 3. Section 4 discusses the accuracy of the ML models and the variation of wind energy resources. A summary of results is presented in section 5.

## 2. Materials and Data

### 2.1 RWP network of China

The RWP is a remote sensing device that can observe the atmospheric wind profiles (Liu et al., 2019). The RWP network of China began to develop as of 2008, and the number of RWP stations developed to 134 by the end of 2020 (Liu et al., 2020). Here, eight RWP stations on the coast from north to south in eastern China are selected, including Dongying, Penglai, Qingdao, Lianyungang, Dayang, Dongtou, Fuqing, and Zhuhai. The spatial distribution of these stations is shown in Fig. 1, marked by red points.

Most stations are located on land along the coast, only Dayang and Dongtou are located on island along the coast (Table 1). The hourly wind speed profiles over the eight stations are obtained from 1 May 2018 to 31 August 2020.

### 2.2 Anemometer

The China Meteorological Administration has established more than 2500 weather stations
instrumented with wind cup anemometers (Mo et al., 2015). The 10-m wind is measured by this wind cup anemometer, which is installed 10 m AGL at the weather station. The sensing part of wind cup anemometer is composed of three or four conical or hemispherical empty cups. It can provide surface wind data with an error of less than 10% (Zhang et al., 2020). Here, the 10-m wind speed data at the eight stations were also obtained from 1 May 2018 to 31 August 2020. The 10-m wind speed data was
processed into hourly average value to match the RWP data.

### 2.3 Radiosonde measurement

The RS measurements provide the profiles of wind speed and wind direction twice a day at 0800 and 2000 LST (Guo et al., 2020; 2021b; Li et al., 2021; Liu et al., 2022). The accuracy of RS wind speed is within 0.1 m/s in the PBL (Guo et al., 2021b). Note that only the station of Qingdao is equipped with
RS during the study period from 1 May 2018 to 31 August 2020.

### 2.4 ERA5 data

The fifth generation European Centre for Medium-Range Weather Forecasts atmospheric reanalysis system (ERA5) is the reanalysis combines model data with observations from across the world into a globally complete and consistent dataset using the laws of physics (Hoffmann et al., 2019). "ERA5
hourly data on single levels from 1959 to present" is a dataset of ERA5, which contains a series of surface parameters. It can provide the surface parameters on a 0.25 x 0.25-degree grid (Hersbach et al.,





2020). Here, nine parameters that may affect the variation of wind speed have been collected, including charnock coefficient (Char), forecast surface roughness (FSR), friction velocity (FV), dew point (DP), temperature (Temp), pressure (Pres), net solar radiation (Rn), latent heat flux (LHF), and sensible heat

flux (SHF). These data were also obtained from 1 May 2018 to 31 August 2020 at eight stations. In addition, the hourly wind data can also be provided by ERA5. The u and v component of wind data at 100 m AGL were also downloaded for wind energy assessment.

### 3. Methods

In this section, the classical PLM method was used to retrieve the $WS_{120}$ based on the surface 10-m

wind speed. Three ML algorithms were then attempted to retrieve the $WS_{120}$. Finally, the method of wind energy evaluation is introduced.

#### *3.1 Power law method*

The PLM method was proposed by Hellman et al. (1914). It assumed that the wind speed below 150 m in the PBL varies exponentially with height. As a result, the wind speed at a certain height has been

typically estimated using the following formulae (Abbes et al., 2012):

$$v_2 = v_1 \times \left(\frac{h_2}{h_1}\right)^{\alpha} \tag{1}$$

where $v_1$ and $v_2$ are the wind speed at height $h_1$ and $h_2$, respectively. The $\alpha$ is the wind shear coefficient, which varies with time, altitude, and location. In actual calculation, the general value of $\alpha$ for coastal topography was set to 0.15 based on former studies (Patel et al., 2005; Banuelos et al., 2010).

#### *3.2 Machine learning algorithms*

Three ML algorithms, including the k nearest neighbor (KNN), support vector machine (SVM) and random forest (RF), are applied to retrieve the $WS_{120}$. For the ML algorithms, one of the most important things is to prepare appropriate characteristic values and accurate reference values as input. Here, the input data include surface wind speed ($WS_{10}$) and direction ($WD_{10}$) from wind cup anemometer at 10

m AGL, wind speed ($WS_{300}$) and direction ($WD_{300}$) at 300 m AGL measured by RWP, and nine surface parameters in ERA5. The reference value is the $WS_{120}$ measured by RS. These values are listed in Table 2. We use 5-fold crossover to train ML models. The specific training process of each model is as follows.

#### *3.2.1 K nearest neighbor*

KNN is one of the ML algorithms, which can be used for regression (Altman, 1992; Coomans et al., 1982). As shown in Fig. 2a. its basic idea is to find the nearest K training samples (inside the gray





circle) in the training dataset based on the distance measurement of a given test sample (orange square), and then make predictions. Therefore, the setting of K value is important to the accuracy of the KNN model. Fig. 3a and 3d show the tuning parameter process for K value. The K value varies from 1-20 with an interval of 1. Correlation coefficient (R) and root mean square error (RMSE) were used to evaluate the accuracy of the model. We need to set an appropriate K value to maximize R and minimize RMSE. According to the curve of R and RMSE changing with K value, the R reach to 0.77 and RMSE is 2.44 m/s when the K was set to 3. Therefore, the K value was set to 3 for KNN model.

*3.2.2 Support vector machine*

SVM is a linear classifier with separation hyperplane with maximal interval (Cortes et al., 1995). As shown in Fig. 2b, the red line and Δ represent the separation hyperplane and edge distance, respectively. The principle of SVM model is obtained a hyperplane with maximum geometric interval to divide the training data set correctly. For SVM model, the penalty parameter (C) is a value that must be specified in advance. C value determines the loss caused by outliers. The loss of objective function is increased with C value when the sum of relaxation variables of all outliers is certain. Therefore, it needs to take an appropriate C to ensure the performance SVM model. As seen in Figs. 3b and 3e, the value of R increases first and then decreases with the increase of C. On the contrary, the RMSE decreases first and then increases with the increase of C. When C equals 0.75, R reaches the maximum value (0.79) and RMSE reaches the minimum value (1.74 m/s). Therefore, the C value was set to 0.75 for SVM model.

*3.2.3 Random forest*

RF model is one of the cluster classification models (Breiman, 2001). As shown in Fig. 2c, the RF model is composed of many decision trees, and each decision tree is irrelevant. For RF model, the number of tree is an important parameter to achieve the optimal performance of the model. Figures 3c and 3f show the tuning parameters process for number of tree (N). The N value varies from 1-500 with an interval of 20. It can find that the R increased with N value increased, while the R was almost unchanged when N value is greater than 100. When N equals 300, R reaches the maximum value (0.81) and RMSE reaches the minimum value (1.64 m/s). Therefore, the N value is set to 300 for RF model.

Figure 4 shows the importance analysis of input variable for three ML model. The input variable with importance lager than 0.1 was marked by red bar. For KNN model, the importance value of $WS_{10}$, FV and Char are 0.3, 0.3, and 0.15, which is much larger than that of other input. For SVM model, the importance value of $WS_{10}$ and FV are larger than 0.1, while the importance values of other input are less than 0.1. For RF model, the importance value of $WS_{10}$, FV and Char are 0.23, 0.14, and 0.13,



respectively. Combined with these results, it found that $WS_{10}$ and FV are mainly input feature for these three models. $WS_{10}$ was the surface wind speed measured by wind cup anemometer. FV is a theoretical wind speed at the Earth's surface which increases with the roughness of the surface. This result confirms that the $WS_{120}$ is mainly affected by the surface wind speed and friction. In addition, the importance value of $WS_{10}$ and FV for KNN model is obviously larger than that of other input. By contrary, for SVM and RF model, although the importance value of $WS_{10}$ and FV is large, the importance value of some input variables is also relatively large with varies from 0.05-0.15. It indicated that the factors such as heat transfer and high-altitude wind speed constraints will also be considered in the inversion process of RF model.

### 3.3 Assessment methods of wind energy

For the obtained $WS_{120}$, a series of indicators need to be used to evaluate wind energy, such as Weibull distribution and wind power density (WPD) (Pishgar et al., 2015). These parameters are commonly used to evaluate the wind energy at a certain station (Fagbenle et al., 2011; Liu et al., 2018).

### 3.4.1 Weibull distribution

The Weibull distribution can calculate the cumulative probability F(v) and probability density f(v) function of $WS_{120}$ in a certain period of time, which are expressed as follows (Chang et al., 2011):

$$F(v) = 1 - exp\left[-\left(\frac{v}{c}\right)^k\right] \tag{2}$$

$$f(v) = \frac{dF(v)}{dv} = \left(\frac{k}{c}\right)\left(\frac{v}{c}\right)^{k-1} exp\left[-\left(\frac{v}{c}\right)^k\right] \tag{3}$$

where $v$ is the $WS_{120}$; k and c are the shape parameter of Weibull, and represent the intensity and stability of wind speed, respectively. Saleh et al. (2012) compared different methods to estimate k and c and pointed out that the moments method is recommended in estimating the Weibull shape parameter. Therefore, we use the moments method to calculate the k and c, which shows as follows (Rocha et al., 2012):

$$k = \left(\frac{\sigma}{\bar{v}}\right)^{-1.086} \tag{4}$$

$$c = \frac{\bar{v}}{\mathcal{T}\left(1+\frac{1}{k}\right)} \tag{5}$$

where $\bar{v}$ and σ are the mean and square deviation of $WS_{120}$, respectively, and Γ is the gamma function, which has a standard form as follows:

$$\mathcal{T}(x) = \int_0^\infty e^{-u} u^{x-1} du \tag{6}$$

### 3.4.2 Wind power density


The WPD is the wind energy per unit area that the airflow passes vertically in unit time, and generally takes the form like (Akpinar et al., 2005):

$$WPD = \frac{1}{2}\rho c^3 \mathcal{T}\left(\frac{k+3}{k}\right) \tag{7}$$

where ρ is the air density, k and c are the shape parameter of Weibull (equ.4 and 5), and Γ is the gamma function (equ.6).

## 4. Results and discussion

The accuracy of four methods is first evaluated by comparing with RS measurements. The
characteristics of $WS_{120}$ were then analyzed based on the results from RF model. Finally, the variation of wind resource was analyzed.

### 4.1 Intercomparison of $WS_{120}$ using different methods

To evaluate the performance of four methods, the estimated $WS_{120}$ of PLM, KNN, SVM and RF were compared. Given that only Qingdao has RS data, the comparison of different methods was conducted
based on the data at Qingdao. Figure 5 shows the comparisons between the observed $WS_{120}$ and the estimated $WS_{120}$ for four methods. RMSE is also displayed on the panel. The R (RMSE) of PLM, KNN, SVM and RF models were 0.79 (2.33 m/s), 0.81 (1.97 m/s), 0.85 (1.52 m/s), and 0.94 (1.00 m/s), respectively. No matter from the R or RMSE results, it shows that the accuracy of ML models is better than that of PLM. This is due to the wind speed in the PBL is affected by turbulence, surface friction
and other factors (Tieleman 1992; Coleman et al., 2021). Simple exponential relationship between target wind speed and $WS_{10}$ is unable to obtain the $WS_{120}$ with high accuracy. Combine with the result in Fig. 4a, the KNN model is mainly based on $WS_{10}$ and FV to build the model. It is the mapping relationship between target wind speed and surface wind speed. This is essentially similar to the principle of PLM. Therefore, the performance of KNN model is slightly improved compared with PLM
method. On the contrary, for SVM and RF models, the R and RMSE between the observed $WS_{120}$ and the estimated $WS_{120}$ are significant improvement. Especially for the RF model, the highest R (0.94) and the smallest RMSE (1.00 m/s) show that the RF model is the best model to retrieve $WS_{120}$. This may be duo to it considers more environmental factors, such as SHF, Char, $WS_{300}$, and $WD_{300}$. These results indicated that considering heat transfer and high-altitude wind speed constraints in inversion
process can improve the accuracy of the model.

To better understand the performance of model, the error analysis of the four methods is conducted based on the $WS_{10}$ and FV. The difference between estimated $WS_{120}$ and observed $WS_{120}$ is shown in





Figure 6. The mean difference between PLM-observed, KNN-observed, SVM-observed, and RF-observed are -1.47, -1.00, 0.01 and 0.01 m/s, respectively. The inversion results of PLM and KNN

models are underestimated relative to the RS observations. By contrast, the mean difference of SVM and RF models is obvious smaller than that of PLM and KNN models. Moreover, it found that the deviation of the PLM and KNN is change with the increase of $WS_{10}$ and FV. It indicated that the stability of the PLM and KNN models need to be improved. The stability of SVM model is better than that of PLM and KNN model, but most of the SVM results are still overestimated when FV is larger

than 0.4 m/s. As for RF model, the deviation is relatively stable and does not change with the increase of $WS_{10}$ and FV. It indicated that the performance of RF is better than other three models. Overall, in terms of stability and accuracy, the RF is the best model to retrieve $WS_{120}$.

### *4.2 Characteristics of wind speed*

Figure 7 shows the monthly and diurnal cycles of $WS_{120}$ at eight stations. For all stations, the seasonal

variation of wind speed is obvious. At Dongying, Penglai, Qingdao, and Lianyungang, wind speed is lager in spring (June to September) and lower in autumn (June to September). Especially at Penglai, there is an obvious low wind speed belt in July and August. By contrast, wind speed is higher in winter (December to February) at Dayang, Dongtou, and Fuqing. As for the Zhuhai stations, wind speed is relatively small throughout the year. These results indicate that the monthly variations of wind speed

are significantly different in different regions. It is because of the differences in monsoon and geographical environment (Durisic et al., 2012). Also shown in Fig. 7 is the diurnal variation. At the land stations like Dongying, Penglai, Qingdao, Lianyungang, Fuqing, and Zhuhai, the wind speed is larger at daytime from 0900 to 1600 LST. The daily cycle of wind speed is mainly affected by the changes of sea-land breeze (Liu et al., 2018). The surface is heated by solar radiation at daytime,

causing turbulence to intensify. Strong turbulence leads to large downward transmission of high-level wind, resulting in high wind speed during the day. After sunset, the surface radiation cools and the air layer tends to stabilize, resulting in a gradual decrease in wind speed. Similar diurnal variations in 10 m wind speed were also observed at three other stations in China (Liu et al., 2013). On the contrary, the wind speed at the Dayang and Dongtou (island stations) is higher at nighttime from 1800 to 2300

LST. This is largely due to the much higher specific heat capacity over ocean compared with over land (Li et al., 2018). The land-ocean thermal condition tends to result in a low wind speed at daytime and a high wind speed at nighttime, particularly in the absence of synoptic-scale forcing.

The histograms of wind speed with corresponding Weibull distributions at eight coastal stations are plotted in Fig. 8. The blue bar and pink lines represent occurrence probability and Weibull distributions,





respectively. The Weibull distribution matches well with the frequency of wind speed at all
        observational stations. Moreover, the shape of the Weibull distributions over these stations can be
        divided into two types. One type is the Weibull distributions at Dongying, Penglai, Qingdao,
        Lianyungang, Dayang, and Dongtou, with a peak probability in medium wind speed (about 6 m/s) and
        a low probability in high and low wind speed. The other type is the Weibull distributions at Fuqing

and Zhuhai stations, with a particularly high probability in low wind speed (about 4 m/s) and a
        decreasing probability as the wind speed increases. Moreover, the k and c values at all eight stations
        are listed in Table 3. The higher c indicates that the wind speed is higher, while the k indicates the
        wind stability (Saleh et al., 2012). The wind resources at Dongying, Penglai, Qingdao, Lianyungang,
        Dayang, and Dongtou are richer than those at Fuqing and Zhuhai.

*4.3 Variation of wind resource*

        Figure 9 shows the diurnal variation of mean wind speed and WPD at all eight stations. The blue and
        red lines are the mean wind speed and WPD, respectively. For each station, the diurnal variation of
        WPD follows the same pattern of mean wind speed. On the whole, two diurnal variation patterns can
        be found. One is for land stations, such as Dongying, Penglai, Qingdao, Lianyungang, and Fuqing.

The hourly mean WPD is larger at daytime from 0900 to 1600 LST with a peak at 1400 LST. This is
        mainly due to the influence of the sea-land breeze (Liu et al., 2018). The other is for island stations,
        such as Dayang, and Dongtou. The hourly mean WPD of these stations remains at a high level at all
        day and is relatively large at nighttime from 1800 to 2300 LST. The urban electricity demand usually
        reaches peaks at around noon in the daytime and in the evening (Hong et al., 2012). This means that

the wind energy at the land and island stations can support the power demand during the noon and
        midnight, respectively. When the demand and the supply achieve a balance, wind energy will be used
        more effectively. In addition, it is worth noting that the mean wind speed and WPD at island stations
        are generally higher than that at land stations, which may be due to the difference in specific heats
        between land and sea. Li et al. (2018) also pointed out that the offshore stations offer more wind energy

than onshore stations.

        Figure 10 shows the monthly variation of mean wind speed and WPD at eight stations. Similar to
        diurnal variation, the monthly variation of WPD in eight stations exhibits the same tend as that of mean
        wind speed. However, the monthly variation of WPD varies by station. The monthly WPD of
        Dongying, Penglai, Qingdao, and Lianyungang is relatively high for the period from March to May,

as compared to the much lower values from August to October. It is likely due to the difference in
        monsoon intensity in winter and summer. The winter monsoon is stronger than the summer monsoon.





This result indicates that the wind source of coastline of Shandong province is more adequate in spring and winter months. By contrast, over Dayang and Dongtou, the monthly WPD is maximums in December, while is low in March and April. Moreover, most of the monthly WPD at Dayang and Dongtou are larger than 200 W/m$^2$. This may be due to these two stations are set up on the island, and the wind energy mainly depends on the sea breeze circulations. As for Fuqing and Zhuhai, the WPD maintain a very low value for every month and remain almost constant.

Figure 11 shows the spatial distribution of seasonal mean wind speed and WPD in the coastal regions of China. On the whole, the spatio-temporal variations of wind speed and wind resource calculated from the RWP observations have good consistency with that of ERA5 data. The maximum mean wind speed of 6.79 m/s occurs at Dayang in summer and the minimum mean wind speed of 4.52 m/s occurs at Zhuhai in autumn. Moreover, the mean wind speed at Dongying, Penglai, Qingdao, Lianyungang, Dayang, and Dongtou is relatively higher than that at Fuqing and Zhuhai for all seasons. It indicates that the wind resources may be richer in the coastal region of northern China. As for the seasonal variation, the mean wind speed at Dongying, Penglai, Qingdao, Lianyungang, and Dayang is the larger in spring and summer than other seasons. For other stations, the largest mean wind speed occurs in winter or autumn. According to National Renewable Energy Laboratory standard (Jamil et al., 1995), the WPD of Qingdao, Dayang, and Dongtou are higher than 200 W/m$^2$ in most seasons, and these three stations could be classified as wind power class II stations. Except for island stations at Dayang and Dongtou, the WPD at Dongying, Penglai, Qingdao, and Lianyungang are much greater than those at Fuqing and Zhuhai, irrespective of seasons. Those results indicated that the wind resources in the Bohai Sea and the Yellow Sea coast are more abundant than those in the South China Sea coast. Furthermore, for the coastal region of Bohai Sea and the Yellow Sea, the wind energy resources are the most abundant in spring while for the East China Sea and the South China Sea coast, the wind energy resources are relatively abundant in summer.

## 5. Summary and conclusions

This study used the ML algorithms to evaluate the wind energy resource at eight coastal stations based on the wind speed profile and surface meteorological data from May 2018 to August 2020. Moreover, the accuracy of PLM, KNN, SVM and RF models was compared based on the correlation and difference between observed WS$_{120}$ and estimated WS$_{120}$. Finally, the wind energy resource was evaluated based on the WS$_{120}$ from RF model.

For the four WS$_{120}$ inversion method, the accuracy of ML models is better than the PLM. It is due to the PLM only depends on the constant α to establish the mapping relationship between surface wind



speed and $WS_{120}$. In fact, the α is not constant and changes with height, time and meteorological conditions. This results in a relatively low accuracy of the PLM method. In contrast, the ML models consider the influence of environmental parameters to improve accuracy, such as FV and Char etc. Moreover, it can be noted that there are also differences in performance between different ML models. The results indicate that the RF model is the best model to retrieve $WS_{120}$, followed by SVM model; last are KNN model. This is caused by different decision strategies of the ML models. The variable importance analysis indicated that the model which can comprehensively consider the influence of most variables has the best performance.

The monthly variation of wind resources varies on the coast of China. The wind resources along the Bohai Sea coast have two peaks approximately in May and October. By contrast, the wind resources along the Yellow Sea coast keeps relatively stable without pronounced peak. As for the coastal regions of East China Sea and the South China Sea, the wind resources increase from January, reach the maximum in June or July, and then decrease until December. In terms of the diurnal variation of wind resources, the WPD over land station has a peak at daytime from 0900 to 1600 LST, while the WPD over island station exhibits peak value at nighttime from 1800 to 2300 LST. This means that the wind energy at the land and island stations can support the power demand during the noon and midnight, respectively. When the demand and the supply achieve a balance, wind energy will be used more effectively. As for the spatial distribution of wind resource, the Bohai Sea and Yellow Sea coast have more abundant wind resources than the East China Sea and the South China Sea. The seasonal variations of wind resources vary on the coast of China. The coast of the Bohai Sea and Yellow Sea has the richest wind resources in spring or autumn, while the coast of the East China Sea and the South China Sea has the richest wind resources in summer.

Our work comprehensively assesses the wind energy resources on the coast of China using the state-of-the-art machine learning algorithm, which provides invaluable information for the development of wind energy industry in the coastal regions of China in the future. However, wind energy assessment is only one part of the efficient utilization of wind energy resources. The cost of wind turbines, topography conditions, environment harm, and other factors also need more attention, which deserves further investigation in the future.

**Acknowledgments**

This work was jointly supported by the National Natural Science Foundation of China (under grants 42001291 and U2142209) and by the Project through the China Postdoctoral Science Foundation (under grant 2020M682485).



**Author Contributions**

The study was completed with cooperation between all authors. Jianping Guo and Liu Boming designed the research framework; Liu Boming and Jianping Guo conducted the experiment and wrote the paper; Xin Ma, Hui Li, Shikuan Jin, Yingying Ma, and Wei Gong analyzed the experimental results and helped touch on the manuscript.

**Conflicts of Interest**

The authors declare no conflicts of interest.

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



**Tables:**


**Table 1** Detailed information of the radar wind profiler observational stations.

| Station Name | Station ID | Longitude (°E) | Latitude (°N) | Altitude (km) | Surface types |
|---|---|---|---|---|---|
| Dongying | 54736 | 118.67 | 37.44 | 11.1 | Land |
| Penglai | 54752 | 120.76 | 37.79 | 60.7 | Land |
| Qingdao | 54857 | 120.23 | 36.33 | 12 | Land |
| Lianyungang | 58044 | 119.24 | 34.54 | 4 | Land |
| Dayang | 58474 | 122.04 | 30.64 | 49 | Island |
| Dongtou | 5876S0 | 121.15 | 27.83 | 71 | Island |
| Fuqing | 58942 | 119.39 | 25.72 | 51.7 | Land |
| Zhuhai | 59488 | 113.2 | 22.07 | 30 | Land |






**Table 2** Summary of the parameters used for machine learning algorithms.

| Type of parameters | Name of parameters | Acronyms | Data sources |
|---|---|---|---|
| Input | Charnock coefficient | Char | ERA5 |
| | Forecast surface roughness | FSR | ERA5 |
| | Friction velocity | FV | ERA5 |
| | Dew point | DP | ERA5 |
| | Temperature | Temp | ERA5 |
| | Pressure | Pres | ERA5 |
| | Net solar radiation | Rn | ERA5 |
| | Latent heat flux | LHF | ERA5 |
| | Sensible heat flux | SHF | ERA5 |
| | Surface wind speed | $WS_{10}$ | Anemometer |
| | Surface wind direction | $WD_{10}$ | Anemometer |
| | Wind speed at 300 m | $WS_{300}$ | RWP |
| | Wind direction at 300 m | $WD_{300}$ | RWP |
| Reference | Wind speed at 120 m | $WS_{120}$ | RS |

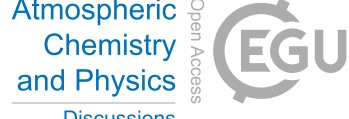

**Table 3** Weibull distribution of $WS_{120}$ on the eight stations from 1 May 2018 to 31 August 2020.

| Station | $WS_{120}$ (m/s) | Standard deviation (m/s) | Weibull Shape factor k | Weibull Scale factor c (m/s) |
|---|---|---|---|---|
| Dongying | 5.54 | 1.77 | 3.46 | 6.16 |
| Penglai | 5.27 | 2.39 | 2.35 | 5.95 |
| Qingdao | 5.86 | 2.45 | 2.58 | 6.59 |
| Lianyungang | 5.81 | 1.75 | 3.68 | 6.43 |
| Dayang | 6.64 | 2.99 | 2.38 | 7.49 |
| Dongtou | 5.89 | 2.66 | 2.37 | 6.65 |
| Fuqing | 5.39 | 2.44 | 2.37 | 6.08 |
| Zhuhai | 4.68 | 1.78 | 2.87 | 5.25 |



**Figures:**


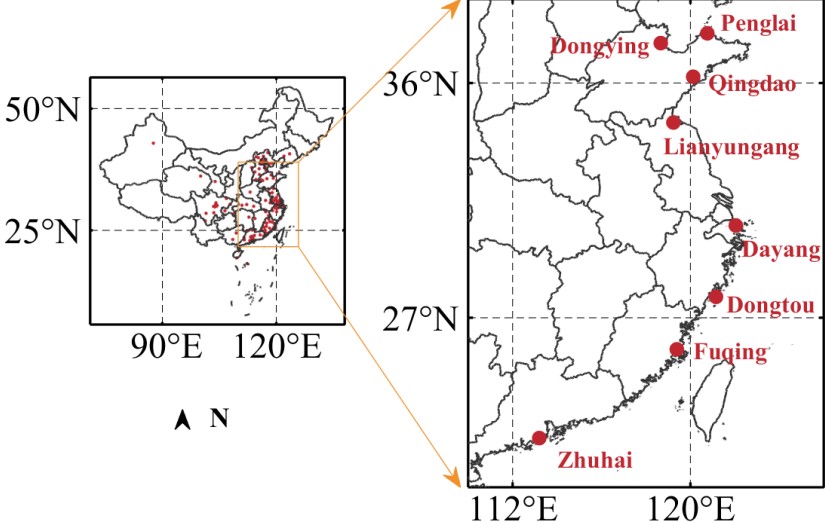

**Figure 1.** Geographical location of the eight radar wind profiler observational stations.






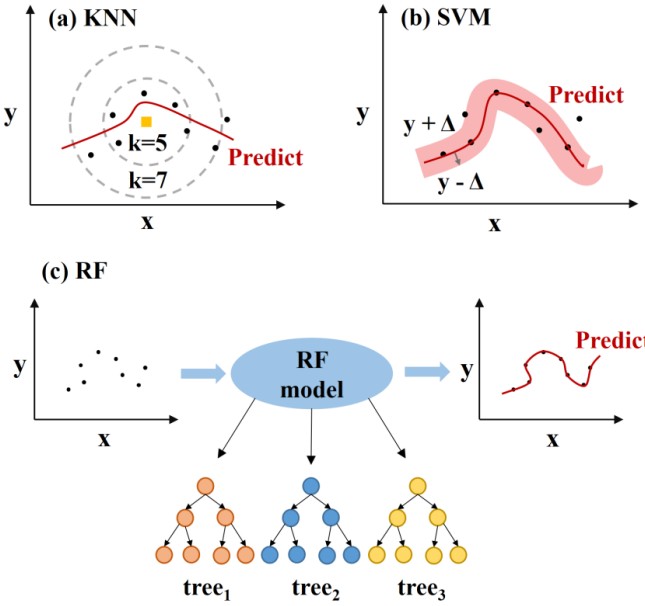

**Figure 2.** Schematic diagram of the (a) K-nearest neighbor (KNN), (b) support vector machine (SVM) and (c) random forest (RF) algorithms used to estimate $WS_{120}$.




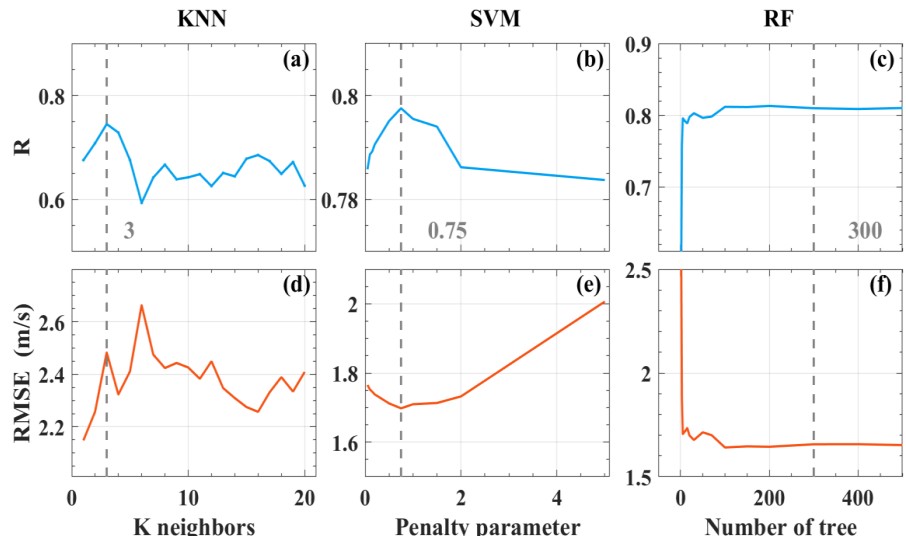

**Figure 3.** The parameter tuning process for (a, d) KNN, (b, e) SVM and (c, f) RF models. The blue and red lines represent the variation of R and RMSE, respectively. The gray dotted line and text

indicate the optimal parameters of the corresponding model.





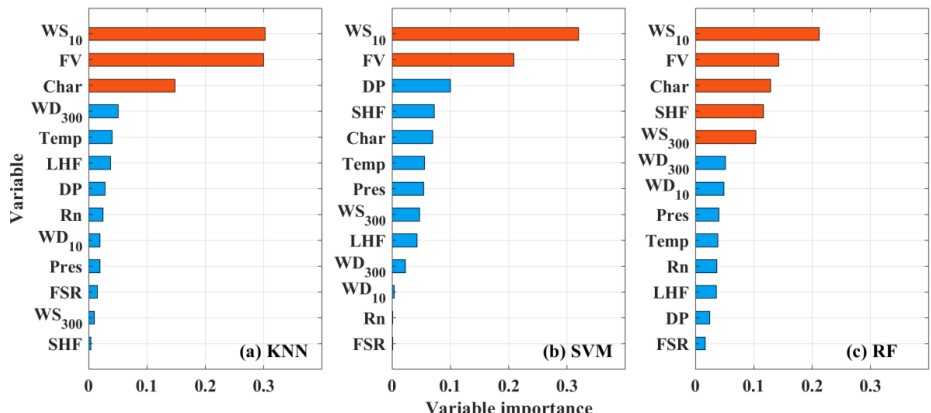

**Figure 4.** Importance analysis of input variables for (a) KNN, (b) SVM, and (c) RF models.



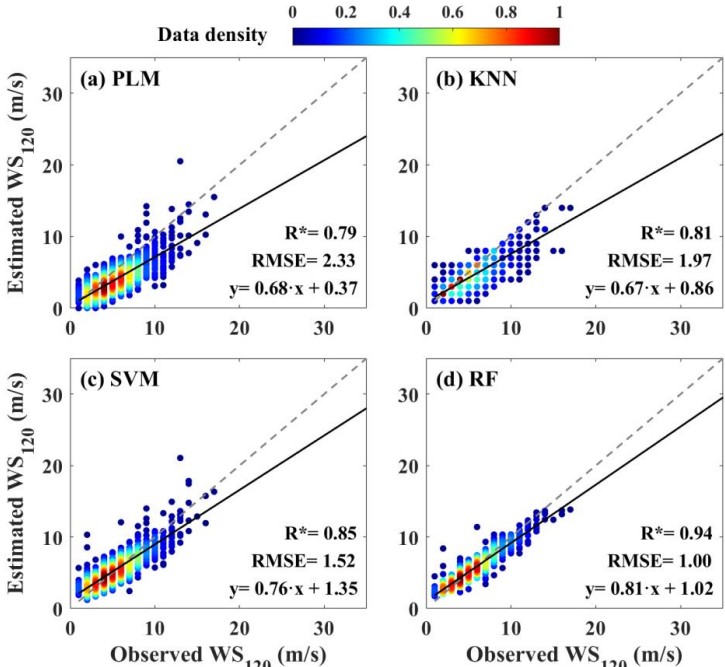

**Figure 5.** Correlation coefficients between observed $WS_{120}$ and estimated $WS_{120}$ based on the (a) PLM, (b) KNN, (c) SVM and (d) RF models. The gray and black line is the reference and regression line, respectively. The asterisk indicates that the correlation coefficient (R) passed the statistical significance difference test (P < 0.05).






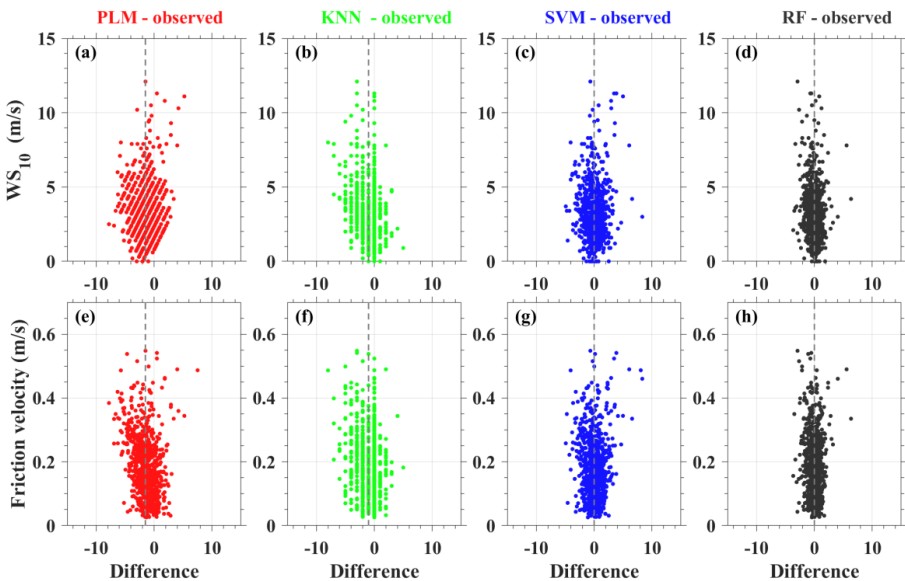

**Figure 6**. Scatter plots showing the difference of observed $WS_{120}$ and estimated $WS_{120}$ as a function of $WS_{10}$ (a-d) and friction velocity (FV, e-h). The red, green, blue and black points represent the difference for PLM-observed, KNN-observed, SVM-observed and RF-observed, respectively. The
gray line represents the mean difference.

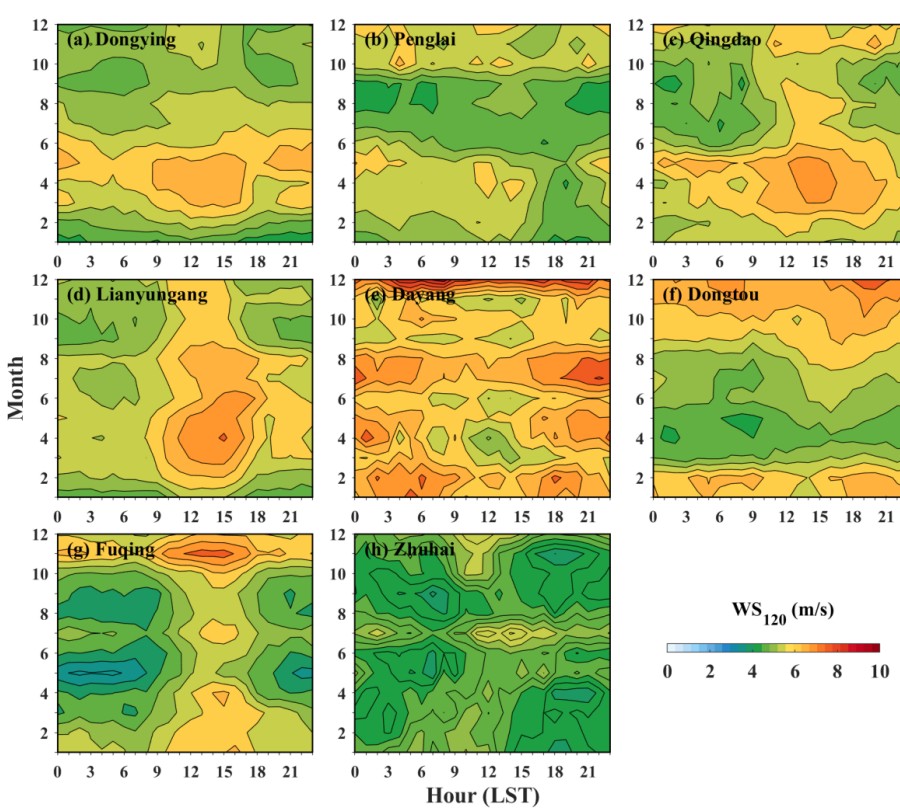

**Figure 7**. Monthly and diurnal cycles of $WS_{120}$ at the eight RWP stations from 1 May 2018 to 31 August 2020.



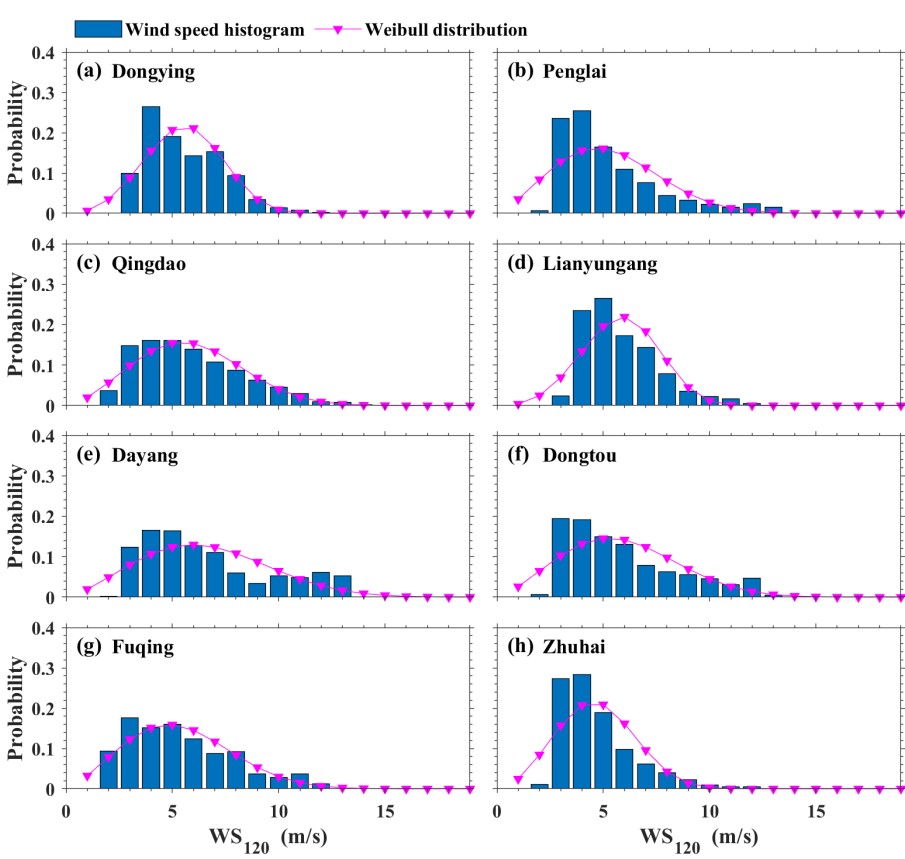

**Figure 8**. Probability distribution and Weibull distribution of $WS_{120}$ at the eight stations from 1 May 2018 to 31 August 2020.





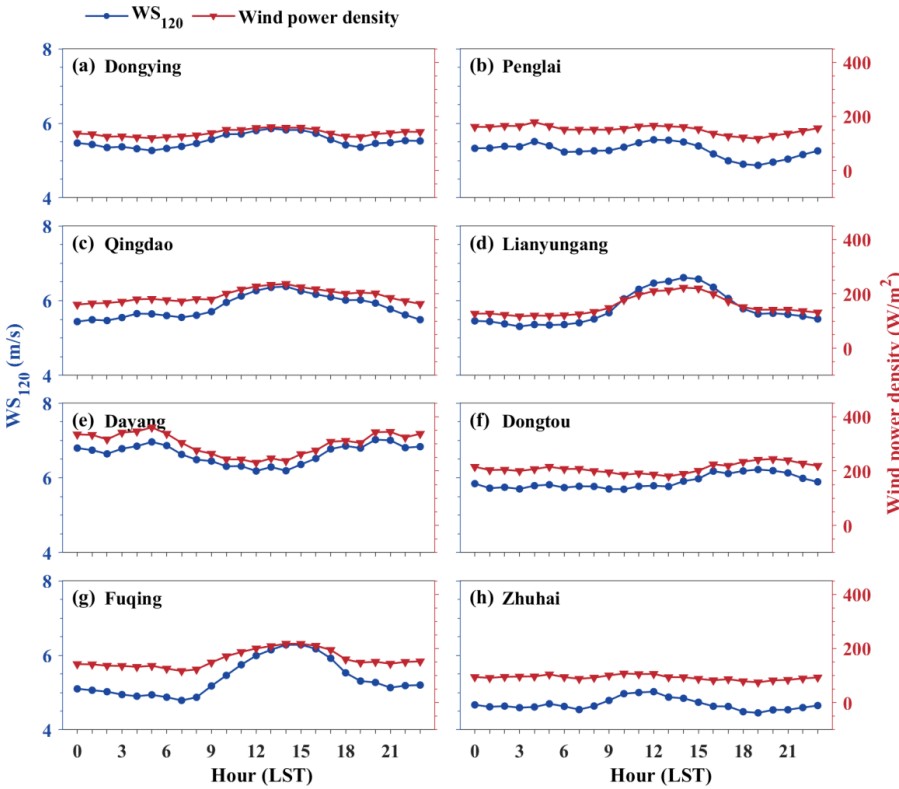

**Figure 9**. Diurnal variation of the $WS_{120}$ and wind power density for the eight stations shown in Fig.

1. The blue and red lines are the mean wind speed and wind power density, respectively.





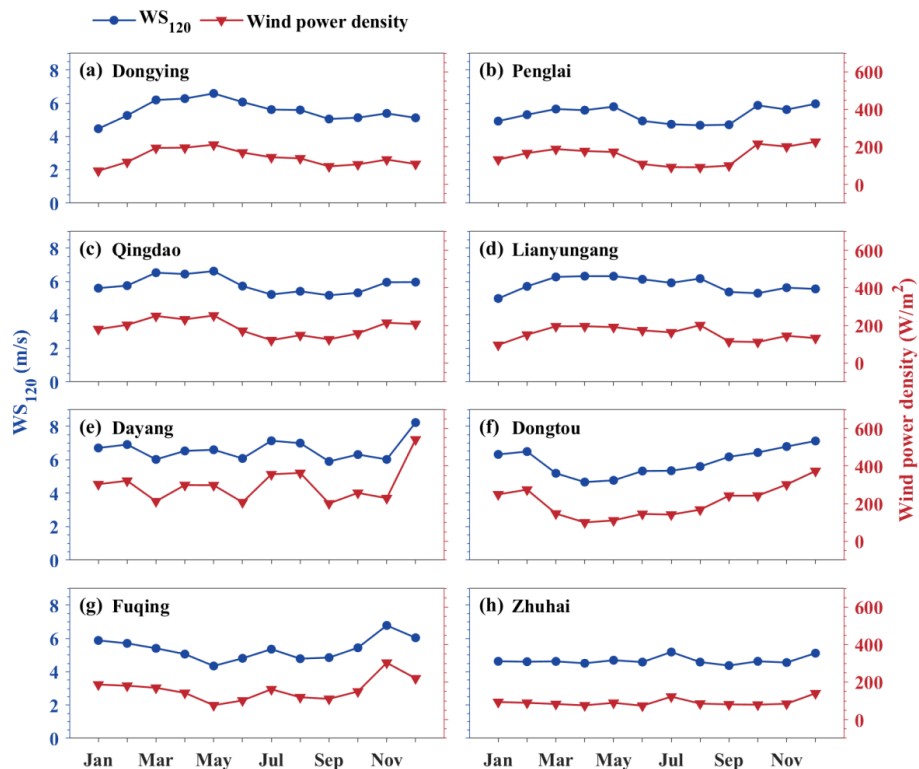

**Fig. 10**. Similar with Fig. 9, but for the monthly variation.




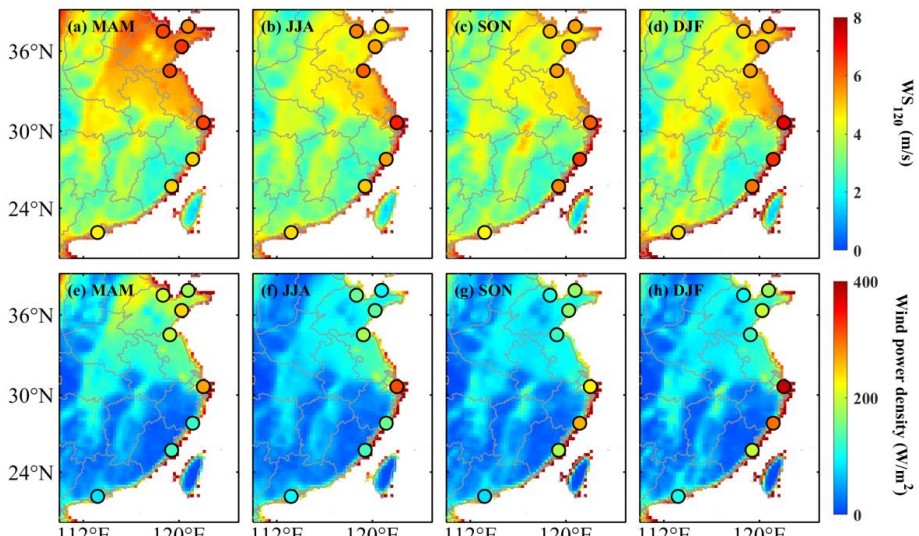

**Fig. 11**. Spatial distribution of the seasonal mean wind speed and wind power density at 100 m AGL along the coastline of China. The circles represent the $WS_{120}$ observations directly from the eight RWP stations. The shading colors in the background show the corresponding results calculated from the ERA5 reanalysis.