# Peer review of "Estimating hub height wind speed based on machine learning algorithm: Implications for wind energy assessment"

_Atmospheric Chemistry and Physics, 2022_

## Author Comment (AC1)

**Response to Reviewer #1's Comments**

Wind energy is one of the most important renewable energy sources in the world. As the author said, China is currently confronted with an increasingly serious energy and climate situation. It is necessary to evaluate the wind energy resources on China. This study attempted to use the machine learning algorithms to evaluate the wind energy resource at eight coastal stations based on the wind speed profile and surface meteorological data. Three ML algorithms and the power law method were used and compared. The wind energy resource is then evaluated based on the wind speed from RF model. Overall, it is a nice and well-organized paper with a clear focus. However, before publishing, there are some problems need to be solved.

***Response: We thank the anonymous reviewer for his/her comprehensive evaluation and thoughtful comments, which greatly improve the quality of our manuscript. We have made great efforts to adequately address the reviewers' concern one by one. For clarity purpose, here we have listed the reviewer' comments in plain font, followed by our response in bold italics.***

1. My biggest concern is whether the reanalysis data (such as ERA5) can be used to evaluate wind energy? In this paper, the author actually uses ML algorithm to predict accurate high-level wind speed, and then conducts wind energy assessment. It means that the accurate high-level wind speed is the most important. As far as I know, ERA5 data can also provide hourly wind speed, so can this data be used? The author should explain this point in the text. Especially in the introduction, the application of reanalysis data is not mentioned at all.

***Response: Good questions! The ERA5 data can provide hourly wind speed, and can also be used to evaluate wind energy. Compared to near-surface in-situ observations or ground-based remote sensing, it has better time continuity and spatial coverage, which can provide data support in the region with poor observational data. But the spatial resolution of the ERA5 data is 0.25 degree \* 0.25 degree.This spatial resolution is much lower than the high-resolution model output such as WRF and the point-based observations. The hourly resolution of ERA5 reanalysis can be used to assess the wind energy in the absence of observational data, which has been reported in previous studies (Laurila et al., 2021; Jurasz et al., 2021; Gualtieri, 2021).***

*According to your suggestion, we added some descriptions in the introduction. "The reanalysis data, such as the fifth generation European Centre for Medium-Range Weather Forecasts atmospheric reanalysis system (ERA5), can provide the hourly wind speed at a specific height (Hersbach et al., 2020; Liu et al., 2020). Compared to near-surface in-situ observations or ground-based remote sensing, it has better time continuity and spatial coverage, which can provide data support in the region with poor observational data. The hourly resolution of ERA5 reanalysis can be used to assess the wind energy in the absence of observational data (Laurila et al., 2021; Gualtieri, 2021). But the maximum spatial resolution of the ERA5 data is 0.25 degree * 0.25 degree. This spatial resolution is much lower than the high-resolution model output such as the weather research and forecasting (WRF) and the point-based observations."*

*References:*

*Laurila, T. K., Sinclair, V. A., & Gregow, H.: Climatology, variability, and trends in near-surface wind speeds over the North Atlantic and Europe during 1979–2018 based on ERA5. International Journal of Climatology, 41(4), 2253-2278, 2021.*

*Gualtieri, G.: Reliability of era5 reanalysis data for wind resource assessment: a comparison against tall towers. Energies, 14(14), 4169, 2021.*

2. P5 Line 140: Why the wind shear coefficient was set to 0.15? Is this value for a site or all sites? What is the uncertainty of this parameter? Some recent publications should be discussed, for example, Retrieval of total and fine mode aerosol optical depth by an improved MODIS Dark Target algorithm; Accuracy, stability, and continuity of AVHRR, SeaWiFS, MODIS, and VIIRS deep blue long-term land aerosol retrieval in Asia; A High-Precision Aerosol Retrieval Algorithm (HiPARA) for Advanced Himawari Imager (AHI) data: Development and verification; Accuracy and error cause analysis, and recommendations for usage of Himawari-8 aerosol products over Asia and Oceania.

*Response: Good questions! In engineering application of power law method, the value of wind shear coefficient is set to a fixed value according to the terrain type (Banuelos et al., 2010). The value of wind shear coefficient is ranged from 0.1 to 0.4 (Li et al., 2018). Here, the general value of α for coastal topography was set to 0.15*

*based on former studies (Patel et al., 2005; Banuelos et al., 2010). Such a setting will inevitably lead to certain errors, but we have not found clear uncertainties in the literature. Therefore, we propose to use machine learning to retrieve wind speed. We added a description to explain the problem. "In engineering application, the value of α is determined by the terrain type, and the variation range is from 0.1 to 0.4 (Li et al., 2018). Here, the general value of α for coastal topography was set to 0.15 based on former studies (Patel et al., 2005; Banuelos et al., 2010)." In addition, the relevant references were also discussed in the manuscript.*

*References:*

*Banuelos-Ruedas F, Angeles-Camacho C, Rios-Marcuello S. Analysis and validation of the methodology used in the extrapolation of wind speed data at different heights. Renew Sustain Energy Rev., 14(8):2383-91, https://doi.org/10.1016/j.rser.2010.05.001, 2010.*

*Li JL, Yu X. Onshore and offshore wind energy potential assessment near Lake Erie shoreline: A spatial and temporal analysis. Energy., 147: 1092-107, https://doi.org/10.1016/j.energy.2018.01.118, 2018.*

*Patel, M. R.: Wind and solar power systems: design, analysis, and operation. CRC press; 2005.*

3. P5 Line 153: The author used a 5-fold crossover to train the model. What is the total number of samples? How to allocate samples for each training?

*Response: Model training is conducted at Qingdao station. A total of 746 sample data are obtained after data matching. 5-fold cross validation means that 80% data is used for training and 20% data is used for testing each time. Repeat five times to traverse all data. We added an explanation to the manuscript.*

4. Fig. 2: This figure is totally unnecessary. KNN, SVM and RF are very classic ML algorithms. It is not necessary to explain its principle. I suggest deleting it or changing it to a supplementary file.

*Response: We change the Fig. 2 to the appendix. At the same time, the description of the ML models has also been modified.*

5. Fig. 11: Similar to the first comment, the ERA5 data can also provide hourly wind speed, so can this data be used to evaluate wind energy? Here, the author used ERA5 data to evaluate wind energy, but did not discuss the application of ERA5 data. If the precision of ERA5 data is high enough, can the ERA5 data be used directly.

*Response: The ERA5 data can provide hourly wind speed, and can also be used to evaluate wind energy. We explained and discussed the application of ERA5 data in the introduction.*

*"Compared to near-surface in-situ observations or ground-based remote sensing, it has better time continuity and spatial coverage, which can provide data support in the region with poor observational data. But the spatial resolution of the ERA5 data is 0.25 degree \* 0.25 degree.This spatial resolution is much lower than the high-resolution model output such as WRF and the point-based observations. The hourly resolution of ERA5 reanalysis can be used to assess the wind energy in the absence of observational data, which has been reported in previous studies (Laurila et al., 2021; Jurasz et al., 2021; Gualtieri, 2021)."*

6. There are many grammatical and spelling mistakes in the text. Please correct them carefully.

*Response: We sought professional organizations to modify the language of the manuscript.*

---

## Author Comment (AC2)

**Response to Reviewer #2's Comments**

The work describes the use of Machine Learning algorithms to evaluate the potential of the use of wind energy in different locations in China. While the results are interesting, there are some revisions that need to be taken before it can be accepted.

***Response: We thank the anonymous reviewer for his/her comprehensive evaluation and thoughtful comments, which greatly improve the quality of our manuscript. We have made efforts to adequately address the reviewers' concern one by one. For clarity purpose, here we have listed the reviewer' comments in plain font, followed by our response in bold italics.***

1. First of all, the Introduction section needs to emphasize better the research gaps that this work aims to fill, with clear and updated references to current literature. It could be interesting to know also if there were similar attempts previously and if yes, how this work is different; if not, why a trial was not made? The rest can be summarized.

***Response: Good suggestion! To our knowledge, few previous studies have attempted to assess the wind energy from the radar wind profiler (RWP) network, which can provide wind profiling much higher than the widely used wind tower or mast. This is the major motivation for our present study. At present, wind energy assessment is mainly based on wind tower, reanalysis data or surface observation data. These methods are widely used in the field of wind energy assessment, but each method has certain limitations (Li et al., 2018; Band et al. 2021). The RWP network of China provides a new data support for wind energy assessment. This is the characteristic of our work. However, there exist large uncertainties in the wind profile observations near the ground surface provided by the RWP, largely due to the influence of ground and intermittent clutter (May and Strauch 1998; Allabakash et al., 2019). Therefore, we attempt to use ML algorithms to fill the gap leaved by the RWP measurements near the ground surface. This is the innovation of this work. As such, we rephrased the related descriptions in introduction section and expanded on the progress in this field by updating the references, as follows:***

***"At present, there are three main methods for wind energy assessment. The first is based on the meteorological tower data (Shu et al., 2016; Liu et al., 2018). The height of the meteorological tower is 100-300 m above ground level(AGL), equipped with anemometer and other meteorological observation instruments. Durisic et al. (2012) analyzed the wind energy at four different heights in the South Banat region based on meteorological tower data. But the construction and maintenance costs of meteorological tower are high, and it is not suitable for large-scale networking observation. The second is based on ground meteorological station data, which can be used to evaluate the wind energy at the hub height by empirical formula (Oh et al. 2012; Liu et al., 2019). Li et al. (2018) investigated the spatial and temporal variations of wind energy near Lake Erie shoreline based on the power law method (PLM). The PLM method generally assumes the wind speed below 150 m in the planetary***

*boundary layer (PBL) varies exponentially with height (Hellman et al. 1914). But due to the influence of inhomogeneous underlying surface, land sea difference and ubiquitous atmospheric turbulence, wind varies constantly and greatly in the vertical (Tieleman 1992; Coleman et al., 2021), posing great challenges and uncertainties to wind energy assessment based on surface observation. The third is based on reanalysis data, such as ERA5. It can provide the hourly wind speed at a specific height (Hersbach et al., 2020; Liu et al., 2020). Compared to near-surface in-situ observations, it has better time continuity and spatial coverage, which can provide data support in the region with poor observational data. The hourly resolution of ERA5 reanalysis has been used to assess the wind energy in the absence of observational data (Laurila et al., 2021; Gualtieri, 2021). But the spatial resolution of the ERA5 data is 0.25 * 0.25 degree, which is much lower than the high-resolution model output such as the weather research and forecasting (WRF) and the point-based observations. These methods are widely used in the field of wind energy assessment (Li et al., 2018; Band et al. 2021), but each method has certain limitations. Therefore, it is necessary to explore more new observation methods to support a comprehensive assessment of wind energy.*

*The radar wind profiler (RWP) network of China can measure the wind profiles from the ground surface to a height of 5-8 km AGL (Liu et al., 2019; Guo et al., 2021a), which provide a novel data source for wind energy assessment. Moreover, increasing wind turbine hub height reduces the impact of surface friction, enabling wind turbines to operate in high-quality wind resource environments (Veers et al., 2019). The RWP can evaluate wind energy at different heights, which is conducive to the selection of wind turbine hub height. Currently, wind turbine is generally installed at the top of wind mast with a height of 100-120 m AGL, which roughly corresponds to the surface layer (Veers et al., 2019). This region is where obstructions such as trees, buildings, hills, and valleys cause turbulence and reduce the wind speed (Stull 1988; Solanki et al., 2022). It leads large uncertainties in the wind profile observations near the ground surface provided by the RWP, largely due to the influence of ground and intermittent clutter (May and Strauch 1998; Allabakash et al., 2019). Therefore, it is necessary to obtain accurate and continuous wind speed at the wind turbine hub height from RWP measurements, which will benefit the robust and scientific assessment of wind energy."*

*We incorporated all the above-mentioned response into the revised introduction part, in which we also updated the literature reviews by referring to the latest references.*

2. As for the methods, the data and instruments used need to be better described. Also, a description of the study site for a non-Chinese could be worthy. The description of the ML methods is not understandable for a non-expert. Limitations of the methods and of the data are never explained.

*Response: According to your suggestions, we made our best to improve the descriptions of data and instruments.*

*Besides, we described the study sites in terms of geographical distribution. These modifications can be seen in section 2.1, as follows:*

*"Here, eight RWP stations on the coast from north to south in eastern China are selected, including Dongying, Penglai, Qingdao, Lianyungang, Dayang, Dongtou, Fuqing, and Zhuhai. The spatial distribution of these stations is shown in Fig. 1, marked by red points. Most stations are located on land along the coast, only Dayang and Dongtou are located on island (Table 1). Geographically, Dongying, Penglai, Qingdao and Lianyungang are located on Shandong Peninsula of northern China, and the other four stations are located on Yangtze River Delta to Pearl River Delta in south China."*

*The descriptions of the ML methods were also rephrased, as per your kind suggestion. These modifications can be seen in section 3.2, as follows:*

*"KNN is one of the simplest ML algorithms, which can be used for regression (Coomans et al., 1982). Its basic idea is to find k nearest neighbors of a sample and assign the average value of these neighbors' attributes to the sample. In this way, the value of the attribute corresponding to the sample can be obtained (Altman, 1992). The schematic diagram of KNN is shown in Fig. S1a. For a given test sample (orange square), it needs to find the nearest K training samples (inside the gray circle) in the training dataset based on the distance measurement, and then assign the average attribute value of the K samples to the test sample. Therefore, the setting of K value is important to the accuracy of the KNN. Here, the KNN algorithm in MATLAB R2020b was used for regression. The code and usage of KNN model are referred to the MATLAB help centre (https://ww2.mathworks.cn/help/stats/fitcknn.html, last access: 15 November 2022)"*

*"SVM is a kind of supervised classification algorithm (Cortes et al., 1995), which can also be used in regression. In regression analysis, SVM is to obtain the optimal fitting curve. The schematic diagram of SVM is shown in Fig. S1b. The red line and Δ represent the fitting curve and slack variable, respectively. The penalty parameter (C) is used to measure the loss caused by outliers. For SVM, it needs to obtain the optimal fitting curve with acceptable loss. The loss of objective function is increased with C value when the sum of relaxation variables of all outliers is certain. Therefore, it needs to take an appropriate C to ensure the performance of SVM. Here, the SVM algorithm in MATLAB R2020b was used for regression. In addition, the code and usage of SVM are referred to the MATLAB help centre (https://ww2.mathworks.cn/help/stats/fitrsvm.html, last access: 15 November 2022)."*

*"RF is an ensemble ML method (Breiman, 2001), which has been widely used in regressive calculation. It is a method to integrate many decision trees into forests and predict the results. Schematic diagram of RF is shown in Fig. S1c. The RF is composed of many decision trees, and each decision tree is irrelevant. The performance of RF is determined by the aggregation of the results of all the trees*

*(Ma et al., 2021). For RF model, the number of trees is an important parameter to achieve the optimal performance of the model. The further detailed information can be referred to Breiman (2001). Here, we used the RF algorithm for regression in MATLAB R2020b. In addition, the code and usage of RF are referred to the MATLAB help centre (https://ww2.mathworks.cn/help/stats/treebagger.html, last access: 15 November 2022)."*

*Limitations of RS, RWP and ERA5 data were also added in manuscript.*

*"One noteworthy drawback is that the operational RS only provide wind profiles twice per day: 0800 and 2000 local solar time (LST)."*

*"Moreover, there exists large uncertainties in the wind profile observations near the ground surface provided by RWP, largely due to the influence of ground and intermittent clutter (May and Strauch 1998; Allabakash et al., 2019)."*

*"But the spatial resolution of the ERA5 data is 0.25 \* 0.25 degree, which is much lower than the high-resolution model output such as the weather research and forecasting (WRF) and the point-based observations."*

*We also discussed the limitations and uncertainties of methods used in our study, by adding the sensitivity analysis, which is mainly reflected in section 3.3. Part of this revision is shown as follows:*

*"To discuss the generalization of the different methods, we investigated the difference between estimated $WS_{120}$ and observed $WS_{120}$ varied with $WS_{10}$ and FV (Fig. 4). Since the model is expected to be applicable to various input values, the variation of the deviation with the input features can reflect the generalization of the model. It found that the deviation of the PLM and KNN is change with the increase of $WS_{10}$ and FV. It indicated that the generalization of the PLM and KNN need to be improved. The generalization of SVM is better than that of PLM and KNN, but most of the SVM results are still overestimated when FV is larger than 0.4 m/s. As for RF, the deviation is relatively stable and does not change with the increase of $WS_{10}$ and FV. It indicated that the generalization of RF is better than other three methods. This is due to RF adds random disturbance in the sample space, parameter space and model space, thus reducing the impact of "cases" and improving the generalization ability (Breiman, 2001). Moreover, it can be seen from the importance analysis that besides WS10 and FV, the RF also depends on Char, SHF and $WS_{300}$. Fig. S2 shows the difference between estimated $WS_{120}$ and observed $WS_{120}$ varied with these three inputs. The deviation is also stable and does not change with the increase of Char, SHF and $WS_{300}$. The results show that the RF has a good generalization to the value changes of all input features. For the other stations, similar coastal environment will not significantly change the input features. Therefore, the RF has sufficient generalization and can be used in other stations. In addition, it notes that the ML model needs to be retrained and set new parameters when using in other environments, such as desert area."*

*All these revisions had been incorporated in this revised manuscript.*

3. Finally, as for the results: the discussion should be improved, citing relevant literature to explain them. An effort must be made to take the discussion to a higher scientific level, now it is limited to a qualitative description of the plots, but reasons for the findings are seldom given, often without citing relevant literature. Sometimes, the description of the processes is also not right, or at least superficial. This for instance applies to the effect of turbulence on wind speed. Or for instance to the factors that need to be taken into account into the WS120 estimation: why are they important? Or why are they not?

*Response: Thanks for pointing these issues out. Per your thoughtful and critical comments, we tried our best to improve the discussion and updated the literatures cited in our revised manuscript. Moreover, more in-depth analysis for the finding had been added, especially on the effect of turbulence on wind speed, which was shown as follows:*

*In section 4.1: "The $WS_{120}$ is affected by turbulence, surface friction and other factors (Tieleman 1992; Solanki et al., 2022). The turbulence caused by inhomogeneous underlying surface can change the wind direction and reduce the horizontal wind speed (Coleman et al., 2021). Especially in coastal areas, the sea land interaction and complex surface types make the variations of near surface wind profiles more complex."*

*In section 4.3: "This daily cycle of $WS_{120}$ is mainly affected by the solar radiation and sea-land breeze. On the one hand, the surface is heated by solar radiation at daytime, warming the low-level air. The convection formed by rising warm air mass results in high wind speed during the daytime. After sunset, the surface radiation cools and the air layer tends to stabilize, resulting in a gradual decrease in wind speed (Liu et al., 2018). On the other hand, the difference of specific heat capacity between sea and land can form the difference of thermal properties between sea and land. The difference of air pressure is obvious, which is easy to form sea land breeze (Li et al., 2020)."*

*Regarding why the myriad factors had been considered into the estimate of WS120, we made the clarification as follows:*

*In section 2.4: "Due to wind is caused by uneven heating of the earth's surface and gradient difference of atmospheric pressure (Solanki et al., 2022). Therefore, nine parameters that may affect the variation of wind speed have been collected, including charnock coefficient (Char), forecast surface roughness (FSR), friction velocity (FV), dew point (DP), temperature (Temp), pressure (Pres), net solar radiation (Rn), latent heat flux (LHF), and sensible heat flux (SHF). Char, FSR and FV are related to surface roughness and friction, and can evaluate the influence of different surface types on the wind speed in the surface layer. DP, Temp and Press are the meteorological parameters associated with wind speed. Rn, LHF and SHF indicate the solar radiation level, which is directly related to the generation of wind."*

*In section 3.2.4: "Combined with these results, it found that $WS_{10}$ and FV are mainly input features for these three models. $WS_{10}$ was the surface 10 m wind speed.*

*FV is a theoretical wind speed at the Earth's surface which increases with the roughness of the surface. This result confirms that the $WS_{120}$ is mainly affected by the surface wind speed and terrain type. In addition, the importance value of $WS_{10}$ and FV for KNN is obviously larger than that of other input. By contrary, for RF, although the importance value of $WS_{10}$ and FV is large, the importance value of some input variables is also relatively large with varies from 0.1-0.15. It indicated that the factors such as heat transfer and high-altitude wind speed constraints will also be considered in the inversion process of RF."*

*All the above response and revisions had been incorporated into section 4 of this revised manuscript.*

4. A strong limitation of the work is that the comparison of observations with model estimations is carried out at a single location, whereas the retrievals are then used at eight different stations. It is not clear if the results obtained at the single station, from which a single ML algorithm was selected, also apply to the other stations, and why.

*Response: Good suggestion! In our opinion, the RF method can be used in other stations. The main reasons are as follows: On the one hand, the RF model has good generalization. The generalization ability is reflected in that the model can give relatively stable output results when inputting different input values (Ma et al., 2021). As shown in Fig. 4, the deviation of RF model is relatively stable and does not change with the increase of $WS_{10}$ and FV, which indicates that the RF has good generalization. It is known that RF adds random disturbance in the sample space, parameter space and model space, thus reducing the impact of "cases" and improving the generalization ability (Breiman, 2001). On the other hand, the similar coastal environment in the other stations will not significantly change the input features. Figure S2 shows the distribution of main input variables of RF model ($WS_{10}$, FV, Char, SHF, and $WS_{300}$) at eight RWP stations. The red dashed lines represent the maximum and minimum values of each variable at Qingdao station. In the range of the red line, the RF can provide stable output due to its good generalization ability. It can be found that almost all the input values of other stations have appeared in Qingdao station. Therefore, the RF model has sufficient generalization and can be used in other coastal stations. In addition, it is noteworthy that the ML model needs to be reconstructed when most of the inputs at a research site are not within the range of the red line.*

*Per your suggestion, we added a section 3.3 to discuss the generalization of the different methods, as follows:*

*"To discuss the generalization of the different methods, we investigated the difference between estimated $WS_{120}$ and observed $WS_{120}$, which as a function of $WS_{10}$ and FV (Fig. 4). Since the model is expected to be applicable to various input values, the variation of the deviation with the input features can reflect the generalization of the model (Ma et al., 2021). It was found that the deviation of the PLM and KNN changed with the increase of $WS_{10}$ and FV. It indicated that the generalization of the*

*PLM and KNN needed to be improved. The generalization of SVM was better than that of PLM and KNN, but most of the SVM results tended to be still overestimated when FV is larger than 0.4 m/s. As for RF, the deviation was relatively stable and did not change with the increase of $WS_{10}$ and FV. This suggested that the generalization of RF was better than other three methods. This could be like due to the fact that RF increased random disturbance in the sample space, parameter space and model space, thereby reducing the impact of "cases" and improving the generalization ability (Breiman, 2001). Moreover, Figure S2 shows the distribution of main input variables of RF model ($WS_{10}$, FV, Char, SHF, and $WS_{300}$) at eight RWP stations. The red dashed lines represent the maximum and minimum values of each variable at Qingdao station. In the range of the red line, the RF can provide stable output due to its good generalization ability. It can be found that almost all the input values of other seven stations have appeared in Qingdao station. Therefore, the RF model is capable of being generalized and can be used in other coastal stations. In addition, it is noteworthy that the ML model needs to be reconstructed when most of the inputs at a research site are not within the range of the red line."*

[Figure]

*Figure S2. The box plot of (a) $WS_{10}$, (b) FV, (c) Char, (d) SHF, and (e) $WS_{300}$ at eight RWP stations. The red dashed lines represent the maximum and minimum values of each variable at Qingdao station.*

5. Line 9: What do you mean by "goal of carbon emission peak"? Revise.

*Response: It means "peak carbon dioxide emissions". As a result, it has been corrected in this revision.*

6. Lines 10-13: The reader may not know what is the "traditional power law method" (and therefore in which sense it relies on the constant coefficient to estimate the high-altitude wind speed) and the variety of factors on which it depends. You should explain better which are those factors or at least some of them.

*Response: Per your suggestion, we rephrased this sentence to "The constant assumption may lead to significant uncertainties in wind energy assessment, given the large dependence on a variety of factors, such as terrain, time and height etc.".*

7. Line 17: Add "of" before "results". I would also add "with the observations" before "show". Change "show" to "shows".

*Response: Amended as suggested.*

8. Lines 18-20: Rephrase: "Based on the WS120 from the RF model, the diurnal variations of WS120 and of wind power density (WPD) were then estimated."

*Response: Amended as suggested.*

9. Line 23: Change "by" to "based on".

*Response: Amended as suggested.*

10. Line 25: This is not the unit for wind speed (could be so for WPD).

*Response: Yes, it is the unit for WPD.*

11. Lines 23-26: But for the reader these names are not meaningful. Please indicate the characteristics of the cities (e.g., geographical location).

*Response: Amended as suggested. "In terms of the spatial distribution of the seasonal mean WPD along the coastal region of China, the WPD at Yangtze River Delta are higher than 200 $W/m^2$ in most seasons, and the WPD at the coastal of Shandong Peninsula and Yangtze River Delta are much greater than at Pearl River Delta."*

12. Line 28: Change "into" to "for".

*Response: Amended as suggested.*

13. Lines 33-34: Well, not only carbon dioxide!

*Response: We modified as: "With the rapid economic development of the world, the massive consumption of fossil fuels produces an increasing emission of carbon dioxide, sulfur dioxide and other pollutants".*

14. Lines 34-35: The link with the previous sentence is missing; and also it is not clear what you mean by "depletion of fossil energy".

*Response: We modified as: "Large amounts of carbon dioxide and other greenhouse gases cause the greenhouse effect, leading to ever-rising air temperature (Shi et al.,2021). To address this problem, it is increasingly becoming imperative to develop renewable clean energy (Hong et al., 2012)."*

15. Line 39: Delete "had".

*Response: Amended as suggested.*

16. Lines 39-40: It would be interesting to know a percentage contribution of this source to the total capacity.

*Response: As shown in the Statistical Review of World Energy(2021), the global wind power generation accounts for about 6% of the total power generation in 2020. To better express the innovation of the manuscript, the introduction had been greatly modified. This descriptions with weak relevance were deleted.*

17. Lines 41-42: The link with the previous sentence is not that clear.

*Response: We deleted this sentence.*

18. Line 54: Perhaps you mean that China is currently facing an increasingly serious energy and climate situation?

*Response: Amended as suggested.*

19. Line 56: Again, I cannot understand what is this "carbon emission peak" strategy.

*Response: It should be the "peak carbon dioxide emissions".*

20. Line 58: Change "has been flourished" to "is flourishing":

*Response: Amended as suggested.*

21. Lines 59-64: This is not linked with the previous sentence. A break is needed.

*Response: We rewrote this paragraph.*

22. Lines 65-66: This is not true as the boundary layer height varies with day, land use, meteorological conditions, and so on. Please revise.

*Response: We rewrote this paragraph and deleted the sentence.*

23. Lines 68-70: You are making a jump to this theory, without introducing the reader to it and to its meaning.

*Response: We rewrote this paragraph and deleted the sentence.*

24. Lines 70-72: Quite generalistic and without details.

*Response: We revised this sentence as follows:*

*"The second is based on ground meteorological station data, which can be used to evaluate the wind energy at the wind turbine hub height by empirical formula (Oh et al. 2012; Liu et al., 2019). Li et al. (2018) investigated the spatial and temporal variations of wind energy near Lake Erie shoreline based on the power law method (PLM). The PLM method generally assumes the wind speed below 150 m in the planetary boundary layer (PBL) varies exponentially with height (Hellman et al. 1914)."*

25.Line 73: Change "was" to "is".

*Response: Amended as suggested.*

26. Line 101: Change "developed" to "increased".

*Response: Amended as suggested.*

27. Lines 100-107 and 109-115 and 117-120 and 122-132: Are these data available somewhere?

*Response: We had added the data acquisition methods in corresponding locations. In addition, the data availability statement is added at the end of manuscript.*

28. Lines 109-115: So are the synoptic (or I assume so based on the statement in the abstract, not repeated here) stations set in the same place as those from the RWP network? This is not clear. The instruments and the data used should be better described.

*Response: Yes, the wind cup anemometer and RWP are located at the same place. We revised this paragraph. "The wind cup anemometer can measure the instantaneous wind speed and is installed at 10 m AGL (Mo et al., 2015). The sensing part of wind cup anemometer is composed of three or four conical or hemispherical empty cups. It can provide surface wind data with an error of less than 10% (Zhang et al., 2020). This device is also installed at eight RWP stations. The 10 m wind speed data can be downloaded in http://www.nmic.cn/data/cdcdetail/dataCode/A.0012.0001.html (last access: 15 November 2022)."*

29. Lines 122-124: This seems more like an advertisement rather than a description. Please provide more details.

*Response: We modified as: "The ERA5 is the reanalysis data combining model data and observations, which provides global, hourly estimates of atmospheric variables (Hoffmann et al., 2019). The horizontal resolution can reach 0.25 * 0.25 degree, and there are 137 vertical levels in vertical direction."*

30. Line 126: Which surface parameters?

*Response: "surface parameters" changed to "a series of surface parameters such as temperature, humidity, pressure and radiation etc.".*

31. Line 126: The spatial resolution is very rough compared to the granularity of point observations.

*Response: Yes, we agreed with your opinion. Therefore, we added some descriptions in the manuscript, as follows:*

*"But the spatial resolution of the ERA5 data is 0.25 * 0.25 degree, which is much lower than the high-resolution model output such as the weather research and forecasting (WRF) and the point-based observations."*

32. Line 130: How did you obtain data at eight stations from gridded data?

*Response: We identified the grid that is closest to each RWP station, and then obtained the gridded data accordingly. To clarify this issue, we rephrase the sentence as follows:*

*"According to the longitude and latitude information of the RWP station, the grid where the RWP station is located is selected and those parameters in the corresponding grid are obtained accordingly."*

33. Lines 127-130: It is not clear how these parameters affect wind speed. Explanations are needed here or somewhere in the manuscript.

*Response: We added a description to explain the problem. "Due to wind is caused by uneven heating of the earth's surface and gradient difference of atmospheric pressure (Solanki et al., 2022). Therefore, nine parameters that may affect the variation of wind speed have been collected, including charnock coefficient (Char), forecast surface roughness (FSR), friction velocity (FV), dew point (DP), temperature (Temp), pressure (Pres), net solar radiation (Rn), latent heat flux (LHF), and sensible heat flux (SHF). Char, FSR and FV are related to surface roughness and friction and can evaluate the influence of different surface types on the wind speed in the surface layer. DP, Temp and Press are the meteorological parameters associated with wind speed. Rn, LHF and SHF indicate the solar radiation level, which is directly related to the generation of wind."*

34. Lines 134-136: Rephrase: "In this section, we introduce firstly the classical PLM method to retrieve the WS120 based on 10-m wind speed measurement. Then, we describe the three ML algorithms used to retrieve WS120. We finally present the method for evaluating wind energy."

*Response: Amended as suggested.*

35. Line 138: Change "assumed" to "assumes".

*Response: Amended as suggested.*

36. Line 139: Change "has been" to "is".

*Response: Amended as suggested.*

37. Line 140: Change "formulae" to "formula".

*Response: Amended as suggested.*

38. Lines 143-144: And what is the value for non coastal locations?

*Response: We added a description to explain the problem. "In engineering application, the value of α is determined by the terrain type, and the variation range is from 0.1 to 0.4 (Li et al., 2018)."*

39. Line 153: Add "presented" before "as follows".

*Response: Amended as suggested.*

40. Lines 155-197: To be honest, if one is not expert in those techniques, the explanation is not well understandable. Figures for this Section could be moved to the Appendix, but the explanation of the methods must be rewritten.

*Response: According to the suggestions, the schematic diagram was moved to the appendix, and the explanation of the methods has been modified. These modifications can be seen in section 3.2.*

41. Lines 236-237 and 237-238: Not clear: revise.

*Response: Amended as suggested. "This is due to the PLM depends on the exponential relationship between $WS_{120}$ and $WS_{10}$. However, the $WS_{120}$ is affected by turbulence, surface friction and other factors (Tieleman 1992; Solanki et al., 2022). The turbulence caused by inhomogeneous underlying surface can change the wind direction and reduce the horizontal wind speed (Coleman et al., 2021). Especially in coastal areas, the sea land interaction and complex surface types make the variations of near surface wind profiles more complex. Simple exponential relationship is unable to obtain the $WS_{120}$ with high accuracy, especially at high wind speed condition."*

42. Line 241: Change "significant improvement" to "significantly improved".

*Response: Amended as suggested.*

43. Line 243: Change "duo" to "due" and "to it considers" to "to the fact that it considers".

*Response: Amended as suggested.*

44. Line 250: Change "By contrast" to "Conversely".

*Response: Amended as suggested.*

45. Lines 248-257: The explanation is not clear: revise. Also, I don't understand the need to discuss the difference (a sort of mean bias) when you were discussing the RMSE and R values. Also, it would be needed to understand if the fitting and comparison of model estimations with observations vary with hour of the day, season, or other factors. Also, the discussion could be improved because for instance from Figure 5 I can observe that: RF model is the best but tends to overestimate small values and underestimate high values; similar discussions also for the other models.

*Response: Good question! It is due to the ML models is a nonlinear fitting of the input variables to obtain the prediction results. Therefore, the generalization ability of the models is reflected in the stability of the results under different input values.*

*This discussion is to explain the sensitivity of the models to input variables and discuss its generalization ability. In this revision, we modified this paragraph and removed it to section 3.3.*

*In addition, according to your suggestions, we added the comparison of model estimations with observations under different time and different season. The modifications can be seen in section 4.1. The discussion in this section was also improved.*

[Figure]

**Figure 5. Comparisons between observed WS$_{120}$ and estimated WS$_{120}$ based on the (a, e, i) PLM, (b, f, j) KNN, (c, g, k) SVM and (d, h, l) RF models under different time. The gray and black line is the reference and regression line, respectively. The colorbar represents the data density. The asterisk indicates that the correlation coefficient (R) passed the statistical significance difference test (P < 0.05).**

[Figure]

*Figure 6. Comparisons between observed WS$_{120}$ and estimated WS$_{120}$ based on the (a) PLM, (b) KNN, (c) SVM and (d) RF models under different season. The red, green, blue and black represent spring, summer, autumn and winter, respectively. The asterisk indicates that the correlation coefficient (R) passed the statistical significance difference test (P < 0.05).*

*"Figure 6 shows the comparisons between the observed WS$_{120}$ and the estimated WS$_{120}$ for four methods under different season. The red, green, blue and black represent the spring, summer, autumn and winter, respectively. The PLM performs best in autumn (R=0.83, RMSE=1.95 m/s) and worst in summer (R=0.72, RMSE=2.37 m/s). The slopes of fitting line at spring, summer, autumn and winter were 0.58, 0.47, 0.72 and 0.8, respectively. It shows that the performance of PLM is affected by seasonal factors. This is due to the wind shear coefficient varies with season (Banuelos-Ruedas et al., 2010). In contrast, the comparison results of ML models are less affected by seasonal factor. The fitting result of KNN at different season is similar except for winter. Similarly, the performance of SVM at spring (winter) is similar to summer (autumn). The slopes of fitting line for SVM at spring, summer, autumn and winter were 0.66, 0.67, 0.8 and 0.82, respectively. As for RF, the fitting result in spring is slightly lower than that in other seasons. The slopes of fitting line at four seasons were ranged from 0.75 to 0.85. This indicates that RF is least affected by seasons. Overall, in terms of stability and accuracy, the RF is the best model to retrieve WS$_{120}$."*

46. Lines 259-260: If it is obvious, why do you need to discuss it?

*Response: We deleted this sentence.*

47. Lines 260-263: Can you explain the reason of those seasonal variabilities?

*Response: Amended as suggested. We discussed the reason of those seasonal variabilities in manuscript. "This is due to the influence of East Asia Monsoon and Mongolian cyclones (Yu et al., 2016). The largescale synoptic systems in China have a relatively high occurrence frequency during the cold season (spring and winter), which result in the higher wind speed than warm season (summer and autumn) (Liu et al., 2019)."*

48. Lines 270-271: The mechanism is much more complicated and also variable because of the presence of complex terrain, buildings, sea-land interfaces.

*Response: Modified as "This daily cycle of WS120 is mainly affected by the solar radiation and sea-land breeze. On the one hand, the surface is heated by solar radiation at daytime, warming the low-level air. The convection formed by rising warm air mass results in high wind speed during the daytime. After sunset, the surface radiation cools and the air layer tends to stabilize, resulting in a gradual decrease in wind speed (Liu et al., 2018). On the other hand, the difference of specific heat capacity between sea and land can form the difference of thermal properties between sea and land. The difference of air pressure is obvious, which is easy to form sea land breeze (Li et al., 2020)."*

49. Lines 292-294: Isn't this obvious?

*Response: We deleted this sentence.*

50. Line 293: Change "On the whole" to "Overall".

*Response: Amended as suggested.*

51. Line 313: Change "maximums" to "maximum".

*Response: Amended as suggested.*

52. Table 1: Are you sure about the unit for Altitude (km)? Some altitude values are really high.

*Response: Sorry. The unit should be "m".*

53. Table 2: This Table is not needed, this explanation can be given in the main text.

*Response: According to the suggestions, this table was moved to the appendix.*

54. Figure 4: Please discuss somewhere the importance of the parameters and the reason.

*Response: The importance of the parameters and the reason were discussed in section 3.2.4.*

*"Figure 3 shows the importance analysis of input variables for three ML models. The importance of the variable indicates the dependence of the model on this parameter. The input variables with importance lager than 0.1 were marked by red bar. For KNN, the importance values of $WS_{10}$, FV and Char are 0.3, 0.3, and 0.15, which are much larger than that of other inputs. For SVM, the importance values of $WS_{10}$ and FV are larger than 0.1, while the importance values of other inputs are less than 0.1. For RF, the importance values of $WS_{10}$, FV and Char are 0.23, 0.14, and 0.13, respectively. Combined with these results, it found that $WS_{10}$ and FV are mainly input features for these three models. $WS_{10}$ was the surface 10 m wind speed. FV is a theoretical wind speed at the Earth's surface which increases with the roughness of the surface. This result confirms that the $WS_{120}$ is mainly affected by the surface wind speed and terrain type. In addition, the importance values of $WS_{10}$ and FV for KNN is obviously larger than that of other inputs. By contrary, for RF, although the importance values of $WS_{10}$ and FV are large, the importance values of some inputs are also relatively large with varies from 0.1-0.15. It indicated that the factors such as heat transfer and high-altitude wind speed constraints will also be considered in the inversion process of RF."*

55. Figure 5: What is the color of the scatterplot representing?

*Response: The colorbar represents the data density. Since some data points overlap, the redder the place, the denser the data. We added a description to the caption.*

56. Figure 7: The plot is quite strange and is not well presented in the main text and in the caption.

*Response: Good suggestion! Considering that we also needed to discuss the daily and monthly changes of wind speed in section 4.3, we deleted this plot and moved the relevant discussion to section 4.3.*

57. Figure 8: The probability distribution looks quite far from the fitted distributions. Please discuss.

*Response: Amended as suggested. We discussed the reason of this probability distribution in manuscript. "In addition, it notes that there is a deviation between the probability density function and the frequency of occurrence at some stations. It is due to the Weibull distribution generally has long tail effect, which also indicates right skewed distribution (Pishgar-Komleh et al., 2015)."*

58. Figure 9 and 10: Please adjust the scales for the y-axes.

*Response: Amended as suggested.*

---

## Author Response (AR2)

**Response to Reviewer #2's Comments**

*We thank the anonymous reviewer for his/her comprehensive evaluation and thoughtful comments, which greatly improve the quality of our manuscript. We have made efforts to adequately address the reviewers' concern one by one. For clarity purpose, here we have listed the reviewer' comments in plain font, followed by our response in bold italics.*

1. First concern relates with the argument of this work, which to me does not seem well related with this journal. As from the journal's homepage, "Atmospheric Chemistry and Physics (ACP) is a not-for-profit international scientific journal dedicated to the publication and public discussion of high-quality studies investigating the Earth's atmosphere and the underlying chemical and physical processes. It covers the altitude range from the land and ocean surface up to the turbopause, including the troposphere, stratosphere, and mesosphere. The main subject areas comprise atmospheric modelling, field measurements, remote sensing, and laboratory studies of gases, aerosols, clouds and precipitation, isotopes, radiation, dynamics, biosphere interactions, and hydrosphere interactions (for details see journal subject areas). The journal scope is focused on studies with important implications for our understanding of the state and behaviour of the atmosphere." Here, the authors have used different machine learning algorithms to investigate the wind energy resource in China, so the connection with the above is not totally convincing. Also because the authors never try to connect or motivate their findings, for instance the construction of the algorithms, with some known physical process and the discussion mostly is narrowed on statistical parameters to compare the results obtained by the algorithms.

*Response: Good point! This concern mainly is caused by our inappropriate description on the grand challenge for the current instruments used to provide wind measurements at turbine height for wind energy industry. To this end, we attempt to obtain accurate wind speed at turbine height based on machine learning algorithms. At present, the wind mast or tower can provide wind speed below 100 m AGL (Durisic et al. 2012; Liu et al., 2018). By comparison, the radar wind profiler (RWP) can measure the wind profiles from the ground surface to a height of 5-8 km AGL (Liu et al., 2019). But there is a large uncertainty in the wind profile observations near the ground surface (below 300 m) provided by the RWP, due to the influence of ground and intermittent clutter (May and Strauch 1998; Allabakash et al., 2019). It leads to a gap (100 to 300 m) in the observation of surface layer wind profile. This height (100~300 m) is also the installation height of the wind turbine. The PLM method is most often applied to extrapolate the surface wind speed to the wind turbine hub height. Previously, a semi-empirical relationship, termed "log wind profile", is commonly used to describe the vertical pattern of horizontal wind speeds within the surface layer. However, a recent study (i.e., Jung et al., 2021) suggested that the error in the wind power density estimation over China can reach to 30 % by taking the*

*above-mentioned relationship. This is largely due to the inconsideration of turbulence nature in the boundary layer, and to the ignorance of the impact induced by multi-scale circulation. To better fit the scope of ACP, we have made substantial changes to the original manuscript.*

*First of all, the title is modified to "Estimating hub height wind speed based on machine learning algorithm: Implications for the wind energy assessment".*

*Secondly, the motivation of this manuscript has been rewritten in Introduction part by highlighting the challenges we are facing in estimating the wind speed at turbine hub height. In this context, machine learning algorithms have been reviewed as well. The updated introduction part goes as follows:*

[revised manuscript text omitted]

*All these revisions had been incorporated in this revised manuscript.*

2. Second point is that although they present limitations coming from the use of ERA5 reanalysis due to its coarse resolution, they have used this dataset for information on local parameters, such as surface roughness, friction, besides wind speed so I do not understand how they can overcome those limitations with their method. In my opinion, at least some of these parameters would need much fine resolution to be representative of the site.

*Response: Good question! As your said, some parameters, such as surface roughness (FSR), friction velocity (FV), temperature (Temp) and pressure (Pres), need much fine resolution to be representative of the site. In the surface layer, the FSR, FV and the atmospheric stability are the main factors controlling the wind profile (Gryning et al., 2007). Unfortunately, there is no field observation data of FV and FSR at Qingdao station. Therefore, we can only obtain the corresponding parameters from the reanalysis data (ERA5). However, it should be noted that for some parameters, such as FSR, Temp and Press, we have used the fine resolution or site observation data to investigate their impact on model accuracy. At land stie, FSR is a measure of surface resistance, which is derived from the vegetation type and snow cover (Smith, 1988). Li et al. (2021) retrieved the FSR in Chinese mainland based on MODIS 0.05 \* 0.05-degree surface type data. They confirm that the FSR at cropland is most likely to 0.3 m. The surface type of Qingdao station is cropland. Therefore, we set the FSR to 0.3 to compare with the FSR from ERA5. In fact, the FSR from ERA5 also approximates a constant value (0.3 m). In addition, the Temp and Press obtained from field observation are also compared with that from ERA5. Fig. S1 shows the influence of different sources of parameters on RF model accuracy. It can find that for these three parameters, the use of field observation or ERA5 data has little impact on the accuracy of the RF model. Therefore, we finally decided to use ERA5 data as input.*

[Figure]

*Figure S1. Comparison of the RF model accuracy under different input. Red and blue points represent the results from ERA5 data and field observation (or fine resolution), respectively.*

*In the revised manuscript, we added a section about the feature selection of RF model. Following the research of Gregori et al. (2022), the inputs, which cannot cause a 2% reduction in correlation coefficient, are regarded as irrelevant feature and removed. The final input variables of RF are $WS_{10}$, FV, Char, SHF and $WS_{300}$. The specific modifications are as follows:*

*"To estimate the $WS_{120}$, $WS_{160}$ and $WS_{200}$, we need to build RF model on 120 m ($RF_{120}$), 160 m ($RF_{160}$) and 200 m ($RF_{200}$), respectively. For each model, it is necessary to select*

*the main features from the inputs to avoid data redundancy and reduce the complexity of the model (Ma et al., 2021). Following the research of Gregori et al. (2022), the inputs, which cannot cause a 2% reduction in correlation coefficient, are regarded as irrelevant feature and removed. Figure 3 shows the importance analysis of inputs for three RF models. The relevant features are marked by red bars. The irrelevant features are marked by blue bars, which are not regarded as final inputs in three RF models. For three RF models, the relevant features are both $WS_{10}$, FV, Char, SHF and $WS_{300}$. It indicates that the factors such as surface friction, heat transfer and high-altitude wind speed constraints are considered in the construction of RF models. In addition, it is surprising that FSR has such low importance in three RF models construction. FSR is a measure of surface resistance, which directly affects the near-surface wind speed (Smith, 1988). At a land station, the FSR is derived from the vegetation type (Li et al., 2021). The surface type of Qingdao station is cropland. Li et al. (2021) confirms that the FSR at cropland is most likely to 0.3 m. In training data, the FSR from ERA5 also approximates a constant value (0.3 m). Since the constant variable has no meaning for RF model construction, the RF model divides FSR into irrelevant variable. Therefore, the final inputs for three RF models are $WS_{10}$, FV, Char, SHF and $WS_{300}$.*"

*All the above response and revisions had been incorporated into section 3.2 of this revised manuscript.*

3. In any case, as regards the revision, the authors have tried to address all the comments from the two reviewers. Grammatical and spelling mistakes are still there, both in the original and in the newly added or modified parts.

*Response: Thanks for pointing these issues out. We tried our best to correct spelling and grammatical errors in the revised manuscript.*

4. In reference to my previous question 4 (A strong limitation of the work is that the comparison of observations with model estimations is carried out at a single location, whereas the retrievals are then used at eight different stations. It is not clear if the results obtained at the single station, from which a single ML algorithm was selected, also apply to the other stations, and why.), I am not satisfied with the answer as it refers to a general statement on the method not specific to this work.

*Response: In the revised manuscript, we focused on the inversion of near-surface wind profile based on the data of Qingdao station. Therefore, this section is changed to analyze the generalization and applicability of RF in Qingdao Station. The specific modifications are as follows:*

*"The accuracy and generalization of the RF model depend on training and testing samples (Ma et al., 2021). However, the training and testing samples are obtained at 0800 and 2000 LST. It needs to discuss whether the RF model also applies to other times. This depends on whether the RF model has enough generalization for the training samples, and whether the inputs at other times have appeared in the training samples. Fig. S3-S5 shows the difference between estimated wind speed and observed wind speed of three RF models, which as a function of the inputs. For three*

*RF models, the deviations are relatively stable and not change with the increase of inputs. It indicates that three RF models have good generalization for the training and testing samples. This is due to the fact that RF tends to increase random disturbance in the sample space, parameter space and model space, thereby reducing the impact of "cases" and improving the generalization ability (Breiman, 2001). Moreover, Fig. S6 shows the distribution of inputs at different time. The red dashed lines represent the maximum and minimum values of each variable at training samples. In the range of the red line, three RF models can provide stable output due to its good generalization ability. It can be found that almost all the inputs have appeared in training samples. Therefore, three RF models have sufficient generalization and can be used at other times."*

5. Also the response to question 5 (What do you mean by "goal of carbon emission peak"? Revise) is not satisfying since it refers to a policy perhaps well known for Chinese but not to the general reader.

*Response: Yes, it refers to a policy in China. The Chinese government proposes to peak its carbon dioxide emissions before 2030 and achieve carbon neutrality before 2060. To facilitate understanding, we deleted this sentence.*

6. Finally, response to question 45 (Lines 248-257: The explanation is not clear: revise. Also, I don't understand the need to discuss the difference (a sort of mean bias) when you were discussing the RMSE and R values. Also, it would be needed to understand if the fitting and comparison of model estimations with observations vary with hour of the day, season, or other factors. Also, the discussion could be improved because for instance from Figure 5 I can observe that: RF model is the best but tends to overestimate small values and underestimate high values; similar discussions also for the other models.) is not convincing enough because it does not explain why there are different effects of the seasonal variability on the various models (some are not affected). Also, it does not focus on the subdiurnal variability.

*Response: According to your suggestion, we have added many discussions and explanations in this section. The specific modifications are as follows:*

[Figure]

*Figure 4. Vertical profiles of the mean wind speed from different methods at (a) all time, (b) 0800 and (c) 2000 LST. Red, black and blue lines represent mean wind*

*profile from RS, PLM and RF, respectively. Corresponding color shading areas represent one standard deviation.*

*"In addition, for both PLM and RF, the retrieved wind profile at 2000 LST is closer to the RS observations. The comparisons between the observed wind speed and the estimated wind speed for PLM and RS under different time is shown in Fig. S7. The fitting results of PLM and RF at 2000 LST are slightly higher than that at 0800 LST. It indicates that the performance of PLM and RF vary with hour of the day. This is due to the wind profile depends not only on the surface friction but also on the atmospheric stratification (Gryning et al., 2007). The surface layer is in an unstable stratification due to heat transfer caused by solar radiation during daytime, while the surface layer tends to stable stratification due to surface radiation cools during nighttime (Yu et al., 2022; Solanki et al., 2022). The $WS_{120}$, $WS_{160}$ and $WS_{200}$ are more vulnerable to the surface turbulence due to the unstable stratification during daytime. Therefore, the performance of PLM and RF at nighttime is better than that at daytime.*

*Figure 6 shows the comparisons between the observed results and the estimated results for PLM and RF under different season. The red, green, blue and black represent the spring, summer, autumn and winter, respectively. At three heights, the performance of PLM is the best in winter and the worst in summer. It shows that the performance of PLM is affected by seasonal factors, which is likely due to the wind shear varying dramatically with season (Banuelos-Ruedas et al., 2010). Pérez et al. (2005) indicates that the surface layer wind speed profile is mainly affected by the convection produced by surface heating in summer. The $WS_{120}$, $WS_{160}$ and $WS_{200}$ affect by the surface due to the unstable stratification, which leads that the PLM performs worst in summer. In contrast, during winter, the surface temperature is generally lower than the air temperature aloft creating a stable inversion (Yu et al., 2022; Liu et al., 2022). The $WS_{120}$, $WS_{160}$ and $WS_{200}$ are disconnected from the surface due to stable stratification. It leads that the PLM performs best in winter. As for RF, although the performance in spring is slightly lower than that in other seasons, the fitting results at four seasons are significantly improved compared with the PLM. This indicates that RF is least affected by seasons. The reason is that the RF model is less subjective than PLM because they are data driven. Overall, in terms of stability and accuracy, the RF is more suitable for estimating wind speed at hub height."*

7. Line 9: Perhaps you mean "goal of reducing carbon dioxide emissions"?

*Response: Yes, amended as suggested.*

8. Line 11: change "the" with "a"

*Response: Amended as suggested.*

9. Lines 40-43: Quite simplicist explanation.

*Response: We have revised the introduction. This sentence has been deleted.*

10. Line 64: What is the "peak carbon dioxide emissions"? It is not clear to the non-Chinese reader.

*Response: Amended as "The Chinese government proposes to peak its carbon dioxide emissions before 2030 and achieve carbon neutrality before 2060 (Pei et al., 2022)".*

11. Lines 82-85: Revise this sentence, there is no principal sentence.

*Response: Amended as "Due to the influence of inhomogeneous underlying surface, land sea difference and ubiquitous atmospheric turbulence, wind varies constantly and greatly in the vertical (Tieleman 1992; Coleman et al., 2021)."*

12. Lines 92-93: WRF is just one of the regional/mesoscale models available. Need to clarify better this aspect and the comparison with ERA5 (based on reanalysis and not on a simulation)

*Response: We have revised the introduction. This sentence has been deleted.*

13. Line 94: Not relevant if you do not specify which are those limitations.

*Response: We deleted it.*

14. Lines 127-134: Geographic information and background are not enough for a non-chinese reader.

*Response: In the revised manuscript, we focused on the inversion of near-surface wind profile based on the data of Qingdao station. Therefore, this section is modified as follows:*

*"The spatial distribution and surface type of this station are shown in Fig. 1. Geographically, Qingdao station is located on the south of Shandong Peninsula and lies to the west of the Yellow Sea. To be more specific, this station is set up in the suburb, surrounded by cropland. The altitude of this station is 12 m above mean sea level."*

[Figure]

*Figure 1. Geographical distribution and surface type of the radar wind profiler observational station at Qingdao.*

15. Line 218: change "need" to "needs."

*Response: Amended as suggested.*

16. Lines 589-595: What about the output data and the codes used in this work?

*Response: Amended as "The output data and codes used in this paper can be provided for non-commercial research purposes upon motivated request (Jianping Guo, Email: jpguocams@gmail.com)".*

---

## Author Response (AR3)

**Response to Reviewer #3's Comments**

The wind profile has significant scientific and practical applications for weather forecasting research and the development of the wind energy industry. This study attempts to evaluate hub height wind speed using the random forest (RF) algorithm based on radar wind profiler and surface synoptic observations at the Qingdao station. The results demonstrate that the hub height wind speed retrieved by the RF model is closer to the radiosonde observation. Additionally, the study analyzes the impact of hub height wind speed retrieved by different algorithms on wind energy assessment. Overall, this manuscript is of great interest to researchers in the atmospheric sciences, but some minor issues need to be addressed before publishing.

***Response: We thank the anonymous reviewer for his/her comprehensive evaluation and thoughtful comments, which greatly improve the quality of our manuscript. We have made efforts to adequately address the reviewers' concern one by one. For clarity purpose, here we have listed the reviewer' comments in plain font, followed by our response in bold italics.***

1. In section 4.1, the author highlights that the performance of the RF model is better at night than during the day. However, the fitting results of PLM and RF at 2000 LST are similar to those at 0800 LST. Please provide an explanation.

***Response: Good question! This is because the wind profile also depends on the atmospheric stratification (Gryning et al., 2007). The surface layer is in an unstable stratification due to heat transfer caused by solar radiation during daytime, while the surface layer tends to stable stratification due to surface radiation cools during nighttime (Yu et al., 2022; Solanki et al., 2022). The hub height wind speeds are more vulnerable to the surface turbulence due to the unstable stratification during daytime. Therefore, the performance of PLM and RF at nighttime is better than that at daytime.***

2. Given the numerous input variables in the model, it is recommended to include a table that explains each variable.

***Response: Good suggestion! We add a table in supplementary to explain the inputs.***

**Table S1.** Summary of the parameters used for machine learning algorithms.

| Type of parameters | Name of parameters | Acronyms | Data sources |
|---|---|---|---|
| Input | Charnock coefficient | Char | ERA5 |
| | Forecast surface roughness | FSR | ERA5 |
| | Friction velocity | FV | ERA5 |
| | Dew point | DP | ERA5 |

|  | Temperature | Temp | ERA5 |
| --- | --- | --- | --- |
|  | Pressure | Pres | ERA5 |
|  | Net solar radiation | Rn | ERA5 |
|  | Latent heat flux | LHF | ERA5 |
|  | Sensible heat flux | SHF | ERA5 |
|  | Surface wind speed | $WS_{10}$ | Anemometer |
|  | Surface wind direction | $WD_{10}$ | Anemometer |
|  | Wind speed at 300 m | $WS_{300}$ | RWP |
|  | Wind direction at 300 m | $WD_{300}$ | RWP |
|  | Wind speed at 120 m | $WS_{120}$ | RS |
| Reference | Wind speed at 160 m | $WS_{160}$ | RS |
|  | Wind speed at 200 m | $WS_{200}$ | RS |

3. In section 2, all the data download links should be moved to the section on data availability.

*Response: Amended as suggested.*

4. The text contains a few grammatical and spelling errors that need to be corrected.

*Response: Thanks for pointing these issues out. We tried our best to correct spelling and grammatical errors in the revised manuscript.*

5. Please confirm if the photos in Figure 1 and Figure 2 involve any copyright issues.

*Response: We add the copyright statement in titles of Figure1 and Figure 2.*